# Navigating Massive Visual Context in Retrieval-Augmented Generation via Multimodal Memory Graph

**Qiuchen Wang** [1]  **Shihang Wang** [2]  **Yu Zeng**  **Qiang Zhang** [2]  **Fanrui Zhang** [2]  **Zhuoning Guo** [3]  **Bosi Zhang** [4]
**Wenxuan Huang** [5]  **Lin Chen**  **Zehui Chen**  **Pengjun Xie** [2]  **Ruixue Ding** [2]

## Abstract

Effectively retrieving, reasoning, and under-
standing multimodal information remains a crit-
ical challenge for agentic systems. Traditional
Retrieval-augmented Generation (RAG) methods
rely on linear interaction histories, which struggle
to handle long-context tasks, especially those in-
volving information-sparse yet token-heavy visual
data in iterative reasoning scenarios. To bridge
this gap, we introduce **VimRAG**, a framework
tailored for multimodal Retrieval-augmented Rea-
soning across text, images, and videos. Inspired
by our systematic study, we model the reason-
ing process as a dynamic directed acyclic graph
that structures the agent states and retrieved mul-
timodal evidence. Building upon this structured
memory, we introduce a Graph-Modulated Vi-
sual Memory Encoding mechanism, with which
the significance of memory nodes is evaluated
via their topological position, allowing the model
to dynamically allocate high-resolution tokens to
pivotal evidence while compressing or discard-
ing trivial clues. To implement this paradigm,
we propose a Graph-Guided Policy Optimization
strategy. This strategy disentangles step-wise va-
lidity from trajectory-level rewards by pruning
memory nodes associated with redundant actions,
thereby facilitating fine-grained credit assignment.
Extensive experiments demonstrate that VimRAG
consistently achieves state-of-the-art performance
on diverse multimodal RAG benchmarks.

## 1. Introduction

Recent advances in Multimodal Large Language Models
(MLLMs) (Bai et al., 2025; Team et al., 2025; Singh et al.,
2025; Team et al., 2023) have fundamentally expanded
the capabilities of multimodal agentic Retrieval-augmented
Generation (RAG) (Cho et al., 2024; Arslan et al., 2024;
Wang et al., 2025a; Yu et al., 2024). With the help of search
engines powered by unified embedding models (Guo et al.,
2025; Li et al., 2026; Meng et al., 2025; Sun et al., 2025b;
Faysse et al., 2024), agents powered by MLLMs can retrieve
and reason over large-scale corpora that contain interleaved
text and images. (Su et al., 2025; Wang et al., 2025b; Yu
et al., 2025b; Jeong et al., 2025; Yeo et al., 2025; Geng
et al., 2025). However, in contrast to text, visual data is
token-heavy and often performs semantically sparse relative
to a specific query (Ma et al., 2024; Tanaka et al., 2023;
Wang et al., 2025c). As memory and context management
strategies have been recognized as effective approaches to
optimize long-context tasks (Zhou et al., 2025; Yu et al.,
2025a; Wu et al., 2025; Xu et al., 2025; Chhikara et al.,
2025; Chen et al., 2024; Ye et al., 2025), transferring this
paradigm to efficiently manage massive visual contexts with-
out losing crucial information is a promising direction.

Inspired by these advances, we introduce **VimRAG**, a
novel framework specifically designed for iterative retrieval-
augmented reasoning via a multimodal agentic memory
paradigm. Our approach is motivated by three critical obser-
vations regarding the challenges of adapting context man-
agement and memory-based methods to multimodal settings:
*(i) Misalignment between action history and context prior.*
There is a fundamental mismatch between the agent's actual
execution history and the reshaped prompt presented to the
model. This structural blindness masks crucial state parame-
ters; specifically in RAG tasks, it leads to repetitive queries
and useless interactions with search engines. *(ii) Incon-
sistency between textual memory and visual observation.*
While compressing visual information into textual memory
significantly improves token efficiency, the inherent loss of
fine-grained details creates a semantic gap, often making the
memory ineffective for verification. *(iii) Insufficient and
inefficient supervision.* Current rejection sampling strate-

---

[1]College of Computing and Data Science, Nanyang Techno-
logical University, Singapore [2]Tongyi Lab, Alibaba Group [3]The
Hong Kong University of Science and Technology (Guangzhou)
[4]City University of Hong Kong [5]The University of Hong Kong.
Correspondence to: Ruixue Ding <ada.drx@alibaba-inc.com>.

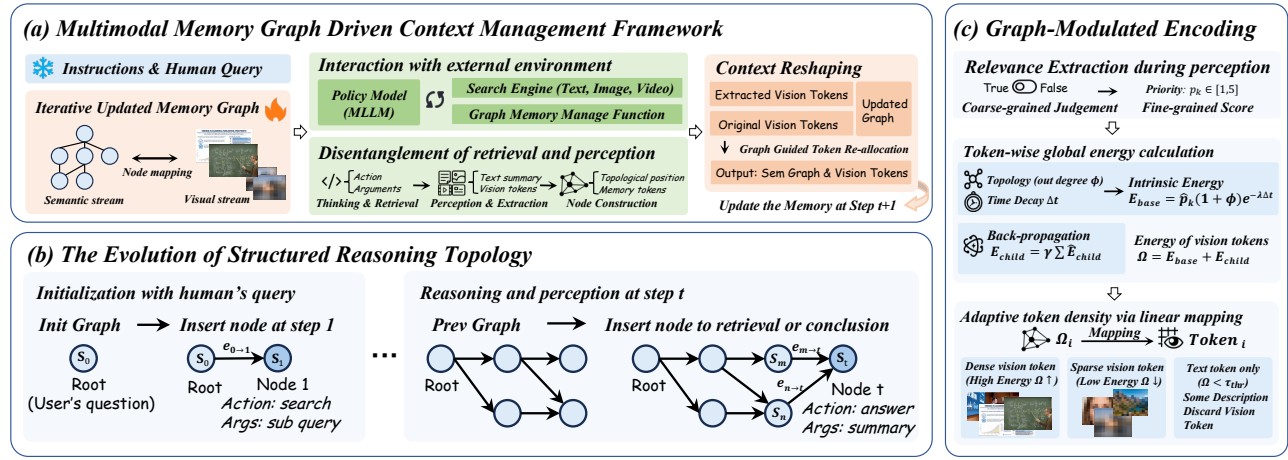

Figure 1. **Inference pipeline of the VimRAG framework.** (a) The cyclic inference loop consisting of reasoning, retrieval, and memory evolution. (b) details the **Evolution of Structured Reasoning Topology**, where each node stores agent-specific memory, including the action, dynamically compressed multimodal observations, and its corresponding temporal and topological structure. (c) illustrates the step-by-step process of **Graph-Modulated Visual Memory Encoding**. This mechanism mimics human forgetting by integrating temporal, topological, and semantic relevance to adjust vision token density, effectively filtering out noise to preserve truly valuable clues.

gies merely broadcast the final outcome-based reward to every step and compute gradients for all response tokens within a long trajectory. This causes misleading supervision in multi-step agentic RAG tasks, where valid retrieval is penalized due to incorrect final answers, while inefficient queries are rewarded solely based on correct outcomes.

To further validate these insights, we conduct a pilot study focusing on three key aspects: the topological organization of interaction history, the trade-off between resolution and efficiency in multimodal memory, and the reliability of supervision based on our multi-step memorization. This leads to our first research question: *1) How should the reasoning process be structured to prevent the loss of critical information during context compression?* Furthermore, the high token cost of preserving visual details raises a second challenge: *2) How can agents resolve the visual memory resolution dilemma under strict token constraints?* Building on these two insights, we pose a third research question: *3) How can we disentangle intermediate interactions from outcome rewards to enable fine-grained supervision?*

In response to these challenges, VimRAG fundamentally reconstructs the agentic reasoning paradigm through three corresponding innovations: (i) To address the structural bottleneck, we propose the *Multimodal Memory Graph*, modeling the reasoning process as a dynamic directed acyclic graph, where each node encodes the agent's action and multimodal observations. As illustrated in Figure 1 (b), this topology captures agent-specific details alongside the temporal and logical dependencies (formalized as state $\mathcal{S}$ and directed edges $\mathcal{E}$). Acting as priors for reasoning, this structure shapes the context for the decomposition of the original question and enables the agent to distinguish between a

*dead-end* branch and a new inquiry, avoiding the potential redundancy found in simple memory appending and the inefficiency of iterative re-summarization. (ii) To shape the context for next-state prediction rather than simply storing facts, we implement a mechanism called *Graph-Modulated Visual Memory Encoding*, built directly upon the graph topology. As shown in Figure 1 (c), by evaluating node significance via topological centrality and recursive feedback, this module adaptively allocates visual token density. It preserves high-resolution tokens for critical evidence while compressing or discarding trivial details, aligning reasoning with valuable observation within a compact token budget. (iii) Observing that the graph topology is naturally suitable for step-wise evaluation, we propose a *Graph-Guided Policy Optimization* strategy for fine-grained supervision. As shown in Figure 4, instead of broadcasting sparse outcome rewards to the samples within the entire trajectory, we utilize the memory graph to perform node pruning by identifying the *critical path* from the root to the answer node. By disentangling the retrieval process from the final outcome, we mask both false positives (irrelevant nodes despite correct answers) and false negatives (valuable retrievals within incorrect answers) during actor updating. This mechanism enhances both training efficiency and effectiveness by focusing gradient updates solely on valid and valuable samples.

Our major contributions are as follows:
• We systematically investigate the multimodal agentic memory paradigm for RAG tasks, identifying critical bottlenecks in context misalignment and supervision sparsity.
• We propose VimRAG, a novel framework that integrates a Multimodal Memory Graph with Graph-Modulated Visual Memory Encoding to structure reasoning topology.

- We introduce a Graph-Guided Policy Optimization strategy to disentangle retrieval validity from sparse rewards, enabling fine-grained credit assignment during training.
- Extensive experiments demonstrate that VimRAG consistently delivers significant improvements, achieving state-of-the-art performance on multimodal RAG benchmarks.

## 2. Pilot Study

In this section, we formulate the problem and investigate challenges of memory paradigms in multimodal RAG, motivating the design of the proposed VimRAG in Sec. 3.

### 2.1. Preliminary

**Task definition.** Given a human query $q$ and a large-scale corpus $\mathcal{C}$ consisting of textual documents, visually rich images, and video streams, our goal is to efficiently retrieve, accurately perceive, and reason over the complex cross-modal information to generate the answer $a$ to query $q$.

**History-Accumulating Paradigm.** Standard agents typically act within a Thought $\tau$, Action $a$, Observation $o$ loop:

$$\mathcal{H}_t = [q, \tau_1, a_1, o_1, \ldots, \tau_{t-1}, a_{t-1}, o_{t-1}] \quad (1)$$

The policy $\pi_\theta(\cdot|\mathcal{H}_t)$ generates the next action with the entire sequence. This leads to significant distraction from critical information $\mathcal{O}_{\text{crit}}$, particularly for sparse multimodal cues. The information density $|\mathcal{O}_{\text{crit}}|/|\mathcal{H}_t| \ll \epsilon$ as $t$ increases.

**Memory-Based Agentic Paradigm.** In contrast, memory-augmented approaches shift from passive history accumulation to active context management. The model updates memory state $m_t$ based on the most recent observation $o_t$:

$$m_{t-1} \xrightarrow{\pi_\theta(\cdot)} (\tau_t, a_t) \xrightarrow{Env} o_t \xrightarrow{\pi_\theta(\cdot|\tau_t, a_t, m_{t-1})} m_t \quad (2)$$

This mechanism maintains a high attention concentration, as the information density remains stable (i.e., $|\mathcal{O}_{\text{crit}}|/|m_t| \approx C$). However, relying solely on the compressed state $m_t$ introduces *Markovian blindness*, leading to potential information loss and disjointed reasoning, which poses challenges for designing a robust memory paradigm.

### 2.2. Impact of Memory Structure on Agentic Reasoning

This subsection investigates the impact of memory structure on the fundamental capabilities of multimodal RAG agents.

**Experimental Settings.** We compare three agentic memory paradigms based on current context management method: *1) Traditional ReAct* (Yao et al., 2022), which simply concatenates the $(\tau, a, o)$ to form the entire context; *2) Iterative Summarization as Memory* (Zhou et al., 2025), which iteratively compresses the observation into the previous memory state; and *3) Structured Graph as Memory*, which maintains a structured topology of agent's reasoning state. We

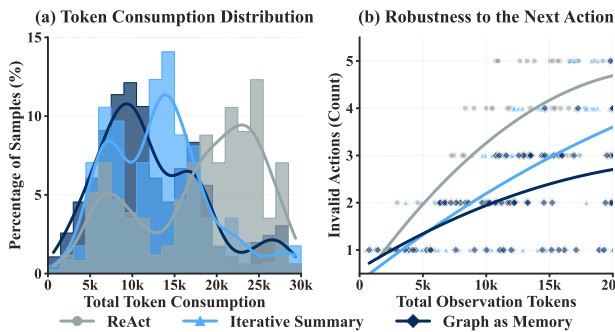

*Figure 2.* **Quantitative analysis of memory structures.** (a) Distribution of total token consumption for complete samples. (b) Count of Invalid Retrieval Action. By modeling the agent's current state rather than just storing facts, the Graph-based paradigm effectively avoids repetitive retrieval compared to the summary-based method.

implement these paradigms by prompting Qwen3VL-30B-A3B-Instruct. See Appendix C.1 for the detailed workflow.

**Observations.** Figure 2 (a) confirms that memory-based paradigms (Summary and Graph) significantly reduce token consumption compared to ReAct. Regarding action robustness (Figure 2 (b)), ReAct degrades most sharply as context expands. The Summary-based approach also suffers from *state blindness*: the agent loses track of its historical retrieval-perception actions, leading to repetitive queries common in multi-hop scenarios. In contrast, by systematically tracking the agent's state, the Graph-based memory effectively minimizes redundant searches and cyclic errors.

**Insights.** The true value of memory lies in shaping future behavior for agents, not merely storing facts from past observations. We argue that a graph-based structure provides the necessary structural bias to maintain the agent's reasoning state, enabling it to bridge valid paths from past to future.

### 2.3. Impact of Information Compression in Memory

This subsection investigates the semantic alignment between observations and memory, as well as the trade-off between compression ratio and critical information preservation.

**Experimental Settings.** We compare four cross-modality compression strategies on the memory paradigm (Zhou et al., 2025): *1) Pre-Captioning:* The visual component of the corpus is pre-captioned. The agent performs purely textual retrieval and memorization. *2) Visual Observation as Memory:* The agent directly stores observations as raw multimodal tokens. *3) Context-Aware Captioning:* The agent retrieves raw multimodal data as observations but memorizes it as textual summaries. *4) Semantically-Related Visual Memory:* After retrieving visual data, the agent selectively retains relevant vision tokens, while irrelevant tokens are discarded. Please see Appendix C.2 for implementation details.

**Observations.** As shown in Table 1, pure text strategy

*Table 1.* **Comparison of cross-modality memory strategies.** Semantically-Related Visual Memory achieves a superior trade-off between compression ratio and critical information preservation.

| Memory Strategy | Modality Retrieval → Memory | Average Tokens | Performance (%) Image | Video |
|---|---|---|---|---|
| (1) Pre-Caption | Text → Text | 0.9k | 14.5% | 17.2% |
| (2) Raw Visual Tokens | Vision → Vision | 15.8k | 45.6% | 30.4% |
| (3) Context-Aware Caption | Vision → Text | 1.5k | 52.8% | 39.5% |
| (4) Semantically-Related | Vision → Selective Vision | 2.7k | **58.2%** | **43.7%** |

*(strategy 1)* minimizes token consumption but suffers from the representation gap between text and vision. Meanwhile, simply storing all raw observations in the context *(strategy 2)* performs poorly due to a decreasing signal-to-noise ratio, consistent with our insights in Section 2.2. The superiority of selective vision *(strategy 4)* over textual summaries *(strategy 3)* demonstrates the necessity of preserving critical visual features for the agent's final verification.

**Insights.** Allocating vision tokens specifically to critical visual details in memory is essential for verification in multimodal tasks, thereby retaining high-value evidence while discarding noise for optimal token efficiency.

### 2.4. Impact of Sparse Reward Signal on Credit Assignment within the Memory Paradigm

This subsection investigates the reliability of outcome rewards for credit assignment in multi-step agent trajectories.

**Experimental Settings.** Let a trajectory be a sequence of steps $\tau = \{s_1, s_2, \ldots, s_T\}$ (Zhou et al., 2025). We decompose steps into two disjoint subsets: *1) Evidence Retrieval* ($\mathcal{S}_{evd}$), containing steps that capture critical clues, and *2) Noise/Redundancy* ($\mathcal{S}_{noise}$), containing irrelevant actions. To evaluate the contribution of specific steps, we conduct a counterfactual ablation study. We construct counterfactual trajectories $\hat{\tau}$ by masking specific subsets of steps and reconstructing the remaining steps into a complete history for re-evaluation. Specifically, for positive samples (where reward $r = 1$), we evaluate $\hat{\tau} = \tau \setminus \mathcal{S}_{evd}$ and $\hat{\tau} = \tau \setminus \mathcal{S}_{noise}$. For negative samples ($r = 0$) that contain valid retrieval steps ($s_t \in \mathcal{S}_{evd}$), we test if performance recovers by denoising, *i.e.*, retaining only the evidence set $\hat{\tau} = \mathcal{S}_{evd}$. Please refer to Appendix C.3 for more details.

**Observations.** Figure 3 (a) illustrates a critical misalignment between the reward $r$ and step-wise samples. Samples with $r = 1$ are not purely efficient; they frequently contain $s_t \in \mathcal{S}_{noise}$ to which outcome-based supervision would incorrectly assign positive gradients. Samples with $r = 0$ should not be universally penalized, as they may contain valid $s_t \in \mathcal{S}_{evd}$ despite the final failure. Figure 3 (b) demonstrates this via counterfactual ablation. For negative samples, simply removing redundant steps recovers performance. This indicates that the failure results from reasoning over noise rather than a lack of evidence. For positive samples, removing evidence steps ($\tau \setminus \mathcal{S}_{evd}$) results

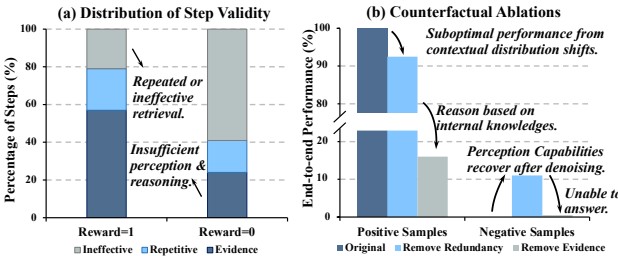

*Figure 3.* **Empirical analysis of misalignment between outcome rewards and step validity.** (a) Distribution of step categories across binary outcome rewards. (b) Impact of removing redundancy or evidence steps, demonstrating the coarseness of rewards.

in a non-zero performance, confirming the model partially relies on parametric internal knowledge.

**Insights.** The memory paradigm naturally decomposes the agentic reasoning process into discrete states, allowing us to disentangle retrieval quality to address the insufficiency of coarse outcome reward. This enables us to calibrate credit assignment, driving the model to learn the distribution of constructive actions at the fine-grained step-level.

## 3. VimRAG

In this section, drawing on the insights and foundational ideas established in the pilot study, we present a comprehensive description of our **VimRAG** framework. We first detail the *Structured Reasoning Topology* (§ 3.1) as the structural backbone of our agentic memory. Next, we introduce *Graph-Modulated Visual Memory Encoding* (§ 3.2) to perform dynamic resolution scaling within memory. Finally, we propose *Graph-Guided Rejection Sampling* (§ 3.3) for fine-grained optimization via topological credit assignment.

### 3.1. Structured Reasoning Topology

We formulate the multimodal reasoning process as the sequential evolution of a directed acyclic graph (DAG), denoted as $\mathcal{G}_t = (\mathcal{V}_t, \mathcal{E}_t)$, where $t$ indicates the discrete reasoning step. Addressing the structural limitations of linear history highlighted in our $1^{\text{st}}$ **insight**, this topology explicitly captures the logical dependencies between agent actions.

**Graph Node as Epistemic State.** We define each node $v_i \in \mathcal{G}_t$ as a discrete unit of agentic epistemic state.

$$v_i \triangleq (p_i, q_i, s_i, m_i), \quad (3)$$

where $v_i$ is defined as a tuple, $p_i$ denotes the set of parent node indices, encoding the local dependency structure; $q_i$ represents the decomposed sub-query associated with the search action; $s_i$ serves as the concise textual summary; and $m_i$ constitutes the multimodal episodic memory bank (e.g., visual tokens from retrieved documents or frames).

The edge set $\mathcal{E}_t = \{(v_j, v_i) \mid j < i\}$ naturally structurally encodes the reasoning flow. The complete graph state is thus an ordered sequence $\mathcal{G}_t = [v_{root}, \dots, v_t]$.

**Iterative Graph Evolution.** We formulate graph construction as a *Partially Observable Markov Decision Process (POMDP)*. At each step $t$, the policy $\pi_\theta$ samples an action $a_t \in \{a^{\text{ret}}, a^{\text{mem}}, a^{\text{ans}}\}$, driving the state transition:

$$a_t \sim \pi_\theta(\cdot \mid \mathcal{G}_{t-1}), \quad \mathcal{G}_t \leftarrow \Psi(\mathcal{G}_{t-1}, a_t), \qquad (4)$$

where $\Psi$ denotes the operator of the external environment.

As shown in Algorithm 1, the model interacts with the external environment in multiple turns within a defined action space. The evolution cycle is categorized into three phases:
- *Exploratory Expansion ($a^{\text{ret}}$):* When current evidence is insufficient, the agent spawns a skeletal node $v'_t = (p_t, q_t, \emptyset, \emptyset)$. The query $q_t$ is executed against an external corpus to retrieve raw multimodal observations $\mathcal{O}_t$.
- *Multimodal Perception & Memory Populating ($a^{\text{mem}}$):* Upon obtaining $\mathcal{O}_t$, the policy invokes the perception action to distill high-entropy information into structured memory: $\mathcal{O}_t \to (s_t, m_t)$. To achieve robust noise suppression, the model follows a coarse-to-fine filtering strategy: for each retrieved item, the model yields a binary saliency mask $u \in \{0, 1\}$ and a fine-grained semantic score $p \in [1, 5]$. For video observations $\mathcal{O}_t^{\text{video}}$, this mechanism leverages the temporal grounding capability of the base model (*e.g.,* Qwen3-VL) to extract keyframes aligned with timestamps. The operation transforms raw data into a summary $s_t$ and visual tokens $m_t$, finalizing the node $v_t = (p_t, q_t, s_t, m_t)$.
- *Terminal Projection ($a^{\text{ans}}$):* Once the policy determines that the reasoning paths within $\mathcal{G}_t$ are sufficient, it executes the answer action. The path from $v_{\text{root}}$ to $v_{\text{ans}}$ constitutes the critical logical and semantic path for task completion.

**Temporally-Grounded Visual Compression.** We leverage the model's temporal grounding capabilities to extract frames of interest, converting from sparse raw observations to dense, semantically-rich representations. The input is represented as an sequence of frames and timestamps:

$$\mathcal{O}_t^{\text{video}} = [(ts_k, f_k)]_{k=1}^n \qquad (5)$$

where $ts_k$ denotes the timestamp (formatted as `<%0.1f seconds>`) corresponding to the $k$-th frame $f_k$. By executing the memory action $a^{\text{mem}}$, the raw stream is distilled into the content of the populated node $(s_t, m_t)$.

### 3.2. Graph-Modulated Visual Memory Encoding

Motivated by **2$^{\text{nd}}$ insight**, we propose *Graph-Modulated Memory Encoding* to address the conflict between visual memory fidelity and token budgets. Instead of static resolution of visual items, we formulate the assignment of vision tokens as a *constrained resource allocation problem*, adaptively allocating high-density tokens to critical evidence.

---

**Algorithm 1** Inference Pipeline of VimRAG

**Require:** Human Query $Q$, Policy $\pi_\theta$, External Environment $\mathcal{V}$.
1: **Initialize:** $\mathcal{G}_0 \leftarrow \{v_{root} : Q\}$, $t \leftarrow 0$.
2: **while** $t < T_{\max}$ **do**
3:      // 1. Context shaping & Action Generation
4:      Context $\mathcal{H}_t \leftarrow \mathcal{V}.\text{LinearizeGraph}(\mathcal{G}_t)$
5:      $a_t \sim \pi_\theta(\cdot \mid \mathcal{H}_t)$
6:      // 2. Topological Expansion (Section 3.1)
7:      **if** $a_t = a^{ret}$ **then**
8:          Initialize empty node $v'_t$ with query $q_t$ and parent $p_t$
9:          Retrieval multimodal information: $\mathcal{O}_t \leftarrow \mathcal{V}.\text{Search}(q_t)$
10:         Multimodal perception: $a^{mem} \sim \pi_\theta(\cdot \mid \mathcal{H}_t, a_t, \mathcal{O}_t)$
11:         Memory population: $(s_t, m_t) \leftarrow \mathcal{V}.\text{Execute}(a^{mem})$
12:         Graph updating: $\mathcal{G}_{t+1} \leftarrow \mathcal{G}_t \cup \{v_t \mid (p_t, q_t, s_t, m_t)\}$
13:      **else if** $a_t = a^{ans}$ **then**
14:          Connect terminal node $v_{ans}$ and **return** $a_t.answer$
15:      **end if**
16:      // 3. Dynamic Visual Memory Shaping (Section 3.2)
17:      **for** each visual node $v_i \in \mathcal{G}_{t+1}$ **do**
18:          Calculate Energy: $\Omega(v_i) \leftarrow \mathcal{V}.\text{Energy}(\mathcal{G}_{t+1})$ (Eq. 7)
19:          Allocate Token Budget: $b_i \leftarrow \mathcal{V}.\text{Scale}(\Omega(v_i))$ (Eq. 8)
20:          Compress Memory: $m_i \leftarrow \mathcal{V}.\text{VisualEncode}(m_i, b_i)$
21:      **end for**
22:      $t \leftarrow t + 1$
23: **end while**

---

**Memory Energy Formulation.** Let $v_i$ denote the $i$-th node in the reasoning graph $\mathcal{G}$, and $\mathcal{M}_i = \{m_{i,k}\}_{k=1}^K$ be the set of $K$ retrieved visual items within this node. As illustrated in Figure 1(c), the energy computation integrates intrinsic priors with recursive reinforcement.

*1) Intrinsic Energy.* First, we compute the baseline energy for each item $m_{i,k}$, denoted as the intrinsic energy $\mathcal{E}_{\text{int}}$:

$$\mathcal{E}_{\text{int}}(m_{i,k}) = \underbrace{\hat{p}_{i,k} \cdot \left(1 + \deg_\mathcal{G}^+(v_i)\right)}_{\text{Structural-Semantic Relevance}} \cdot \underbrace{\exp\left(-\lambda(T - t_i)\right)}_{\text{Temporal Decay}}, \quad (6)$$

where the normalized $\hat{p}_{i,k} \in [0, 1]$ represents the fine-grained semantic priority and $\deg_\mathcal{G}^+(v_i)$ denotes the out-degree of node $v_i$. To mimic human forgetting, temporal decay is applied based on the elapsed time $T - t_i$, thereby preventing the accumulation of outdated information.

*2) Recursive Reinforcement.* Relying solely on intrinsic energy is insufficient, as early evidence often serves as a crucial bridge to downstream insights despite low initial significance. To address this credit assignment problem, we calculate the final energy $\Omega(m_{i,k})$ by reinforcing the intrinsic energy with feedback from successor nodes:

$$\Omega(m_{i,k}) = \mathcal{E}_{\text{int}}(m_{i,k}) + \gamma \sum_{v_j \in \text{Child}(v_i)} \overline{\Omega}(v_j), \qquad (7)$$

where $\gamma$ controls the feedback strength, and $\overline{\Omega}(v_j)$ aggregates the average energy of $\text{Child}(v_i)$. This formulation ensures that foundational nodes supporting high-value future reasoning are preserved against temporal decay.

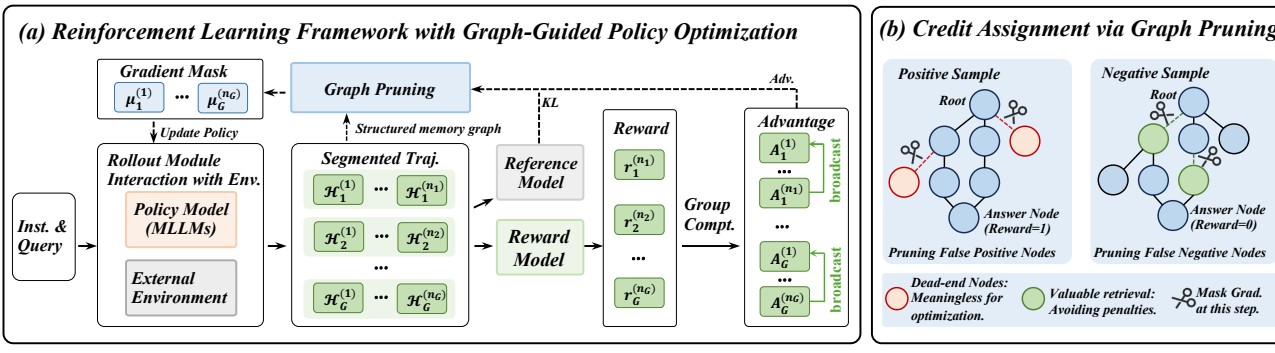

*Figure 4.* **Overview of Graph-Guided Policy Optimization. (a) Agentic Memory Training Framework** segments rollout trajectories into atomic reasoning cycles within the memory paradigm, where outcome-based advantages are broadcasted to enable step-level credit assignment. **(b) Credit Assignment via Graph Pruning** leverages the structured graph for precise credit assignment, applying gradient masks to avoid reinforcing inefficient dead-ends in positive samples and prevent penalizing valuable retrievals in negative samples.

**Global Selection and Resolution Allocation.** As shown in Algorithm 1, we dynamically allocate tokens to each visual item $m_{i,k}$ based on its energy during memory shaping:

$$b_{i,k} = \left\lfloor S_{\text{total}} \cdot \frac{\Omega(m_{i,k})}{\sum_{m' \in \mathcal{M}\text{top}} \Omega(m')} \right\rfloor. \quad (8)$$

where $S_{\text{total}}$ represents the total token budget that maintains optimal model performance, and $\mathcal{M}_{\text{top}}$ denotes the set of top-$K$ items retained based on energy ranking. This formulation ensures that the ViT encoder captures fine-grained details of high-energy evidence, concentrating the computational budget on the most informative visual regions.

### 3.3. Graph-Guided Policy Optimization

Motivated by the **3$^{\mathrm{rd}}$ insight**, we propose the Graph-Guided Policy Optimization (GGPO), where we leverage the graph structure to disentangle reasoning paths and enable fine-grained credit assignment as illustrated in Figure 4.

**Trajectory Segmentation.** We first formalize the initial prompt $\mathcal{C}_t$ at step $t$. It contains the system instruction $inst$, the user's query $q$, and the linearized memory graph $\mathcal{L}(\mathcal{G}_t)$:

$$\mathcal{C}_t = \{inst, q, \mathcal{L}(\mathcal{G}_t)\} \quad (9)$$

During the interaction rollout, the agent collects a structured trajectory decomposed into node-construction units. Each unit corresponds to the construction of a graph node $v_t$:

$$\mathcal{H}^{(t)} = (\mathcal{C}_t, \tau_t, a_t^{ret}, o_t, \tau_t', a_t^{mem}) \to v_t \quad (10)$$

where $\tau_t$ represents the reasoning process leading to the retrieval action, and $\tau_t'$ denotes the reflection used to synthesize the memory action. Each episode concludes with a terminal reasoning block $\mathcal{H}^{(T)} = (\mathcal{C}_T, \tau_{ans}, a^{ans})$.

**Credit Assignment via Graph Pruning.** Misalignment between trajectory-level rewards and the effectiveness of

individual steps is a key challenge. As shown in Figure 4, we address this by leveraging the semantic graph topology to evaluate each step and performing pruning:

*1) Pruning False Positives (Dead-End States).* Given a positive sample $(\mathcal{T}, r = 1)$, we identify the critical path $\mathcal{P}_{ans} \subseteq \mathcal{G}$ by traversing backwards from the answer node. Nodes $v \notin \mathcal{P}_{ans}$ represent *dead ends*, which are redundant explorations or are not logically related to solution.

*2) Pruning False Negatives (Valuable Retrieval).* Given a negative trajectory $(\mathcal{T}, r = 0)$, leveraging the reference annotations for each query, we identify steps where the retrieval results contain relevant information. We exclude these *valuable retrieval* actions from the negative policy gradient update to avoid penalizing effective behavior.

**Implementation of Reinforcement Learning** To implement structural credit assignment, we conduct a binary pruning mask $\boldsymbol{\mu} = [\mu_1^{(1)}, \dots, \mu_G^{(n_G)}]$ for each segmented trajectory, where $\mu = 1$ indicates that the step should be excluded from the update. Let $\mathcal{P}_{ans}$ denote the critical path nodes in a correct solution, and $\mathcal{R}_{val}$ denote nodes yielding valuable retrieval within an incorrect solution. We define $\mu_t$ as:

$$\mu_t = \underbrace{\mathbb{I}(r=1) \cdot \mathbb{I}(v_t \notin \mathcal{P}_{ans})}_{\text{Dead-Ends in Positive}} + \underbrace{\mathbb{I}(r=0) \cdot \mathbb{I}(v_t \in \mathcal{R}_{val})}_{\text{Valuable Retrieval in Negative}} \quad (11)$$

where $\mathbb{I}(\cdot)$ is the indicator function. The first term masks redundant steps in positive episodes, while the second masks valid retrieval actions in negative episodes to avoid penalizing them. The optimization objective is formulated as:

$$\max_{\pi_\theta} \mathbb{E}_{q \sim \mathcal{D}, \{\mathcal{H}_g^{(i)}\}_{g=1,i=1}^{G,n_g} \sim \pi_\theta} \left[ \frac{1}{\sum_{g=1}^G n_g} \sum_{g=1}^G \sum_{i=1}^{n_g} (1 - \mu_{g,i}) \cdot \right.$$
$$\left. \min\left( r_{g,i}(\theta)\hat{A}_{g,i}, \mathrm{clip}(r_{g,i}(\theta), 1-\varepsilon, 1+\varepsilon)\hat{A}_{g,i} \right) \right] \quad (12)$$

where $\{\mathcal{H}_g^{(i)}\}_{i=1}^{n_g}$ represents the sequence of $n_g$ segments corresponding to the $g$-th rollout, and $r_{g,i}(\theta)$ denotes the probability ratio for segment $i$ in rollout $g$.

*Table 2.* **Main Results.** The best performance are marked in bold. The benchmark are divided into three categories based on the modality of the reference: general text, images and visual documents, and large-scale long-context video corpus. The evaluation is conducted on a unified large-scale multimodal dataset, introducing greater challenges and aligning more closely with real-world applications.

| METHOD | GENERAL TEXT | | IMAGE & VISUAL DOCUMENT | | | LARGE-SCALE LONG-CONTEXT VIDEO CORPUS | | | | OVERALL |
|---|---|---|---|---|---|---|---|---|---|---|
| | HotpotQA | SQuAD | WebQA | SlideVQA | MMLongBench | LVBench | WikiHowQA | SyntheticQA | XVBench | |
| *Qwen3-VL-4B-Instruct* | | | | | | | | | | |
| Vanilla RAG | 63.6 | 63.0 | 45.3 | 44.8 | 15.2 | 12.0 | 11.0 | 32.6 | 27.2 | 35.0 |
| ReAct | 62.9 | 63.8 | 39.3 | 44.9 | 13.9 | 12.2 | 14.1 | 29.9 | 21.3 | 33.6 |
| UniversalRAG | 50.9 | 64.7 | 41.2 | 14.7 | 5.4 | 15.5 | 3.3 | 23.9 | 7.5 | 25.2 |
| VideoRAG | 57.2 | 63.2 | 41.8 | 34.0 | 16.2 | 19.4 | 20.1 | 45.8 | 29.8 | 36.4 |
| MemAgent | 67.3 | 70.4 | 46.5 | 43.1 | 12.4 | 18.9 | 19.2 | 34.3 | 24.4 | 37.4 |
| Mem1 | 70.8 | 67.5 | 44.1 | 49.8 | 27.1 | 16.8 | 14.3 | 42.9 | 31.9 | 40.6 |
| **VimRAG (Ours)** | **75.1** | **73.7** | **47.6** | **52.8** | **28.1** | **22.8** | **21.8** | **51.0** | **34.2** | **45.2** |
| *Qwen3-VL-8B-Instruct* | | | | | | | | | | |
| Vanilla RAG | 64.0 | 64.2 | 48.1 | 48.5 | 16.2 | 14.8 | 15.7 | 37.0 | 29.7 | 37.6 |
| ReAct | 70.8 | 65.5 | 40.0 | 50.0 | 15.4 | 15.9 | 23.0 | 35.0 | 24.0 | 37.7 |
| UniversalRAG | 55.9 | 65.3 | 45.8 | 17.2 | 6.6 | 19.1 | 12.0 | 25.0 | 9.6 | 28.5 |
| VideoRAG | 62.0 | 62.2 | 42.1 | 35.5 | 18.2 | 23.8 | 25.7 | 49.5 | 30.7 | 38.9 |
| MemAgent | 71.1 | 74.8 | 47.1 | 45.3 | 14.7 | 22.2 | 23.1 | 37.5 | 26.9 | 40.3 |
| Mem1 | 73.0 | 68.4 | 44.5 | 55.7 | 32.6 | 22.4 | 19.9 | 43.4 | 32.2 | 43.6 |
| **VimRAG (Ours)** | **79.1** | **76.4** | **53.9** | **62.4** | **33.4** | **24.5** | **29.7** | **54.5** | **37.1** | **50.1** |

*Table 3.* **Ablation study.** We decompose the memory paradigm into Memory Structure and Memory Shaping for ablation.

| MEMORY STRUCTURE | | | MEMORY SHAPING MECHANISM | | | Accuracy |
|---|---|---|---|---|---|---|
| Iterative Summary | Graph Topology | Multimodal Nodes | Standard Update | Graph Evolution | Energy-Based Allocation | |
| ✓ | | | ✓ | | | 43.6 |
| | ✓ | | | ✓ | | 47.1 |
| | ✓ | ✓ | | ✓ | | 48.9 |
| | ✓ | ✓ | | ✓ | ✓ | **50.1** |

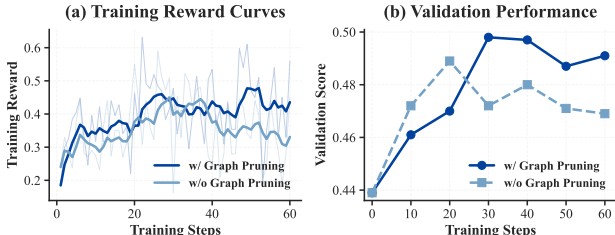

*Figure 5.* **Ablation study on our GGPO**, shows that our method is more robust than the baseline GSPO without pruning, as it updates the actor using distribution-consistent step-wise samples.

# 4. Experiments

## 4.1. Experimental Settings

**Baselines.** We compare our method with current advanced RAG and Context Management methods: (1) **Vanilla RAG** uses the original question as a query for the search engine, then MLLMs perform direct inference. (2) **ReAct** (Yao et al., 2022): The model performs reasoning and retrieving in the think-then-act paradigm. (3) **VideoRAG** (Jeong et al., 2025) performs frame selection to extract the information for inference. (4) **UniversalRAG** (Yeo et al., 2025) introduces RAG within cross-modal corpora as a routing problem. (5) **MemAgent** (Yu et al., 2025a): We implement it by sequentially feeding in the search results. (6) **Mem1** (Zhou et al., 2025) updates its memory through a cyclical retrieval-then-memorization process.

**Benchmarks and Metrics.** We evaluate our method on a comprehensive set of benchmarks covering diverse tasks: general text **HotpotQA** (Yang et al., 2018) and **SQuAD** (Rajpurkar et al., 2016); image-based **WebQA** (Chang et al., 2022); visually rich document benchmarks **Slide-VQA** (Tanaka et al., 2023) and **MMLongBench** (Ma et al., 2024); the long-video benchmark **LVBench** (Wang et al., 2025c); and video corpus understanding benchmarks **Wiki-HowQA** and **SyntheticQA** (Jeong et al., 2025). In addition, we construct **XVBench** to address the lack of benchmarks

for cross-video understanding. We employ a binary model-based metric (0 or 1) to evaluate the correctness of the agent's answer across these tasks. Please refer to Appendix E for more details about dataset and environment setups.

## 4.2. Results

**Main Results.** As presented in Table 2, traditional paradigms such as ReAct struggle with context exhaustion caused by token-heavy visual data. Meanwhile, multimodal RAG methods such as VideoRAG and UniversalRAG, tailored to specific tasks, often exhibit limited generalization due to their fixed inference pipelines. Furthermore, consistent with the state blindness observed in our Pilot Study (Sec. 2), current summarization-based memory paradigms fail to track historical retrieval actions, leading to redundant reasoning loops. By addressing these structural limitations, VimRAG effectively manages massive multimodal contexts and outperforms recent baselines like MemAgent and Mem1. Specifically, VimRAG achieves substantial improvements on both Qwen3-VL-8B-Instruct ($43.6 \rightarrow 50.1$) and Qwen3-VL-4B-Instruct ($40.6 \rightarrow 45.2$). This confirms that explicitly modeling the reasoning topology, rather than passively accumulating history, is essential for unlocking the full potential of MLLMs in multimodal intensive tasks.

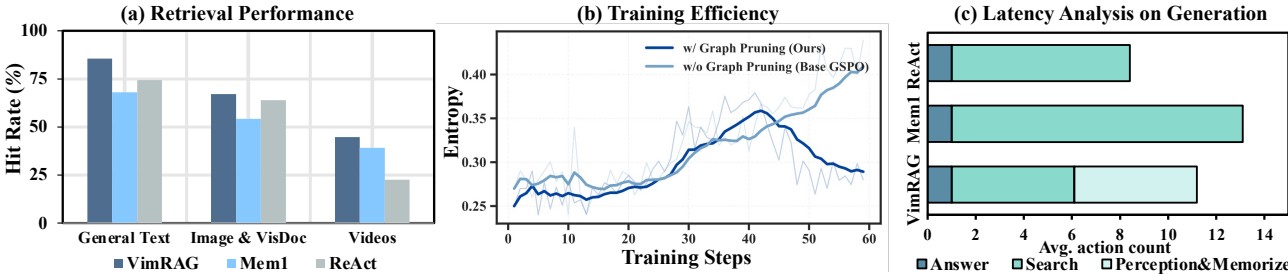

*Figure 6.* **Analysis of Robustness and Efficiency.** (a) Retrieval Hit Rate across modalities. (b) Training entropy curves, demonstrating faster convergence with Graph Pruning. (c) Breakdown of inference steps, highlighting VimRAG's reduced redundancy.

**Approach Ablations.** As shown in Table 3, we decompose key components of VimRAG to examine their impact. The sequential improvements upon introducing modules validate the power of our paradigm. The substantial gain from Graph Topology confirms that explicitly modeling logical dependencies mitigates state blindness, which **proves the necessity of structured state tracking in long-horizon reasoning**. Furthermore, visual memory with Energy-Based Allocation achieves higher accuracy by prioritizing high-resolution tokens for critical nodes, which **proves the effectiveness of our Graph-Modulated Visual Memory Encoding** in optimizing the trade-off between detail and efficiency. Finally, consistent with the stability shown in Figure 5, our Graph-Guided Rejection Sampling disentangles retrieval validity from outcome rewards, which **proves the importance of fine-grained topological credit assignment** for robust training. Overall, Graph as Memory not only structures the agent's reasoning trajectory but also facilitates superior model optimization through structural disentanglement.

### 4.3. Analysis

**Robust retrieval serves as the foundation for high-quality generation.** High-quality generation relies heavily on the precision of the retrieved context. As illustrated in Figure 6 (a), there is a significant performance gap in retrieval robustness between different memory paradigms. Traditional linear or summarization-based methods often suffer from state blindness, where agents lose track of historical execution paths as the context expands, leading to repetitive queries and redundant interactions with search engines.

**Necessity of Memory Modality Alignment.** The results in Section 2.3 reveal that memory modalities better align with the corpus type. Our study confirms that retaining relevant visual tokens significantly outperforms text compression. VimRAG addresses this by adaptively preserving high-resolution tokens for key evidence while discarding noise. This efficient handling of visual information contributes to the superior performance shown in Table 2.

**Structural Disentanglement Accelerates Policy Optimization.** Figure 6 (b) demonstrates that our strategy achieves faster convergence compared to the baseline. This observation yields two critical insights into agentic RL: (1) the stability of optimization depends on ensuring the correctness of positive gradients while eliminating ambiguous updating from negative samples; and (2) the quality of rollout samples are crucial, utilizing samples with clear preference alignment is more decisive for performance and training efficiency than simply scaling up the training set.

**Generation Latency.** Figure 6 (c) reveals that VimRAG significantly reduces the overall trajectory length compared to ReAct and Mem1 despite introducing a perception step. Linear methods often exhibit a heavy tail of token usage caused by repetitive re-reading and invalid searches. In contrast, the structured memory in VimRAG prevents redundant loops and converges to solutions with fewer total actions.

**Case Study.** The case study in Appendix G highlights how VimRAG successfully identifies a dead-end node $v_1$ and backtracks to a new query node $v_2$. This qualitative analysis further validates that our graph topology empowers the agent with human-like self-correction capabilities.

## 5. Related Work

**Multi-modal Retrieval-augmented Generation.** Current Retrieval-augmented Generation method demonstrates significant advantages in addressing knowledge-intensive problems (Riedler & Langer, 2024; Fang et al., 2025; Wang et al., 2025b; Chen et al., 2024; Arslan et al., 2024; Geng et al., 2025; Bonomo & Bianco, 2025; Han et al., 2025; Asai et al., 2024). With the development of multimodal embeddings, building unified multimodal RAG agents has become a mainstream trend (Guo et al., 2025; Yu et al., 2024; Li et al., 2026; Faysse et al., 2024; Meng et al., 2025). Currently, more research applies RAG to challenging tasks like long video understanding and document understanding, extending beyond simple text-based QA (Fan et al., 2024; Jeong et al., 2025; Wang et al., 2025a; Zeng et al., 2025b;a; Shi et al., 2024; Wang et al., 2025d; Li et al., 2024b;a). Our

work builds on these developments to realize a RAG system with interleaved text, image, and video, unifying retrieval, perception, and understanding in one framework.

**Context Management and Memory for Agents.** The most widely used approach for context management in LLM-based agents is ReAct's append-all-history strategy (Yao et al., 2022). As the demand for long-context reasoning grows, researchers are increasingly focusing on optimizing context management (Xu et al., 2025; Zhou et al., 2025; Yu et al., 2025a; Chhikara et al., 2025; Wu et al., 2025; Ye et al., 2025; Chen et al., 2024; Sun et al., 2025a; Zhang et al., 2025). However, visual data is often token-heavy and sparse, requiring more efficient processing methods compared to text. Our approach introduces a structured memory graph to handle these visual features, effectively solving the redundancy problem while preserving critical details.

## 6. Conclusion and Future Work

We propose VimRAG, which uses a dynamic memory graph to handle massive visual contexts. This approach allows fine-grained perception to critical information, leading to better performance in complex multimodal tasks. For future work, we will try our best to train a unified model for multi-task and multi-modal reasoning.

## Impact Statement

This work significantly advances the capabilities of multimodal agentic systems by addressing the critical challenge of navigating massive visual contexts in Retrieval-Augmented Generation. By introducing a structured memory graph and energy-based token allocation, our framework not only enhances reasoning accuracy but also substantially improves computational efficiency, promoting more sustainable deployment of Multimodal Large Language Models in resource-constrained environments. In addition, explicit modeling of inference paths enhances the interpretability and reliability of agent behavior, laying a solid foundation for developing trustworthy multimodal AI systems capable of handling long-horizon tasks.

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

## A. Dataset Construction

The overall data construction pipeline is illustrated in Figure 7. We construct our initial video corpus $\mathcal{V}$ from the Howto100M dataset. To ensure data diversity, we explicitly balance the distribution of video durations. Given a long video $v \in \mathcal{V}$, we divide it into a sequence of segments $\{s_1, s_2, \ldots, s_n\}$. A key feature of our approach is the variable time interval between selected segments. Specifically, we sample segments with both short and long time gaps. This strategy captures dense local contexts and sparse global relationships, respectively. Next, we employ an MLLM to generate detailed captions $C$ for these segments. Based on $C$, the LLM synthesizes a specific query $q$ and the corresponding reasoning steps. Then, we apply a semantic filter to the generated queries. Crucially, this filter ensures that $q$ depends on the large multimodal corpus rather than being a general question. The specific prompt design is provided in Appendix 16. In addition, we sample a subset of the generated data to construct a benchmark, named XVBENCH. This benchmark addresses the lack of evaluation standards for cross-video understanding within large-scale corpus.

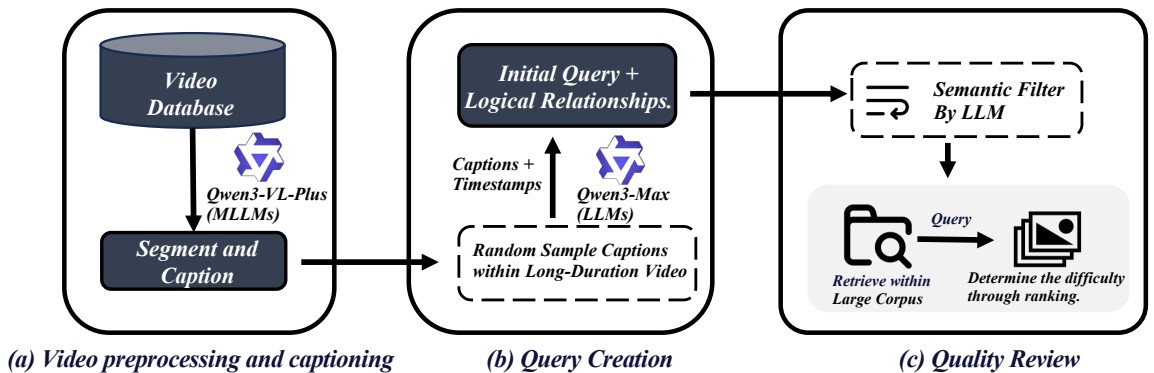

*(a) Video preprocessing and captioning*    *(b) Query Creation*    *(c) Quality Review*

*Figure 7.* **Overview of the data construction pipeline.** The process consists of three stages: (a) **Video Preprocessing**, where long videos are segmented and captioned by MLLMs; (b) **Query Creation**, where LLMs generate complex queries and logical steps based on sampled captions; and (c) **Quality Review**, involving semantic filtering and difficulty ranking to ensure data quality and challenge levels.

## B. Environment and Experimental Settings

**Training and Inference Setups.**    We performed SFT with LoRA (Hu et al., 2022) using Llama-Factory (Zheng et al., 2024) , and RL using rLLM. All experiments were conducted on NVIDIA H20-3e 141G GPUs. To optimize training efficiency for long-context multimodal tasks, we utilized a gradient accumulation strategy with a total batch size of 64. The SFT phase spanned 3 epochs with a learning rate of $1.0e - 4$, while the RL phase employed a more conservative learning rate of $1.0e - 6$ for the actor model to ensure stable policy convergence . Please refer to Appendix F for the detailed hyperparameters.

**The construction of a search engine.**    During training and inference, we built a search engine based on a corpus of approximately 200k multimodal items. The detailed composition of the dataset is shown in Table 4. We use GVE-7B (Guo et al., 2025) as the embedding model because it supports text-to-text, image, and video retrieval. It allows us to measure the distance between items based on their embeddings. For videos, we split them into 1-minute clips and then generate embeddings, following the model's best practices.

*Table 4.* **Statistics of the Corpus.** Our setting merges reference information from different modalities into a unified corpus, which poses a greater challenge for RAG systems and better aligns with real-world applications.

| Dataset | Domain / Category | Corpus Scale | Query Type |
|---|---|---|---|
| HotpotQA | General Text | 3-10 paragraphs per question | Multi-hop reasoning |
| SQuAD | General Text | Single/Multiple passages | Span extraction |
| WebQA | Image-Text | Web-scale snippets & images | Multi-hop multimodal |
| SlideVQA | Visually Rich Doc | 52k+ slide images | Multi-hop & Numerical |
| MMLongBench | Visually Rich Doc | 6,492 documents | Long-context understanding |
| LVBench | Long-context Video | 103 long videos (~68m avg.) | Temporal grounding & Reasoning |
| WikiHowQA | Large Video Corpus | ~500 videos (HowTo100M subset) | Retrieval & Generation |
| Synthetic QA | Large Video Corpus | ~500 videos (HowTo100M subset) | Retrieval |
| XVBench | Cross-Video | Fine-grained Segments form HowTo100M | Cross-video reasoning |
| **Merged (Ours)** | Interleaved Multimodal RAG | ~200k multimodal items containing text, images, and videos | Complex Long-context Interleaved Reasoning |

**Model-Based Reward.** We employ a model-based reward to evaluate the quality and relevance of generated responses. Specifically, we utilize Qwen3-Max (Yang et al., 2025) as our reward model. The prompt used for the reward model is illustrated in Figure 12 (Appendix J.1). Given the input query, reference answer, and generated response, the reward model assesses the correctness of the generated response and outputs a binary value (0 or 1) to represent the accuracy of the answer.

## C. More Details about the Pilot Study

### C.1. Experimental Settings of Memory Structure

To evaluate the impact of different memory structures on agentic reasoning and context management, we conduct a pilot study on a representative subset of the multimodal corpus. In this setting, we specifically construct the search engine using only the video corpus to better observe the interaction between agentic states and action sequences.

We implement and compare three structures: **(1) Traditional ReAct** follows the standard Thought-Action-Observation loop as detailed in Appendix J.4, where the entire interaction history is concatenated linearly. **(2) Iterative Summarization as Memory** continuously compresses observations into the previous memory state as described in Appendix J.5 to maintain context window efficiency. **(3) Structured Graph as Memory** maintains a dynamic directed acyclic graph as shown in Appendix J.6, where each node explicitly stores the decomposed query and its corresponding extracted semantic summary.

The results confirm that ReAct suffer from state blindness, leading to repetitive queries and useless interactions as the context expands. By preserving the agent state through a structured topology that tracks every query and the corresponding information extraction, the graph-based memory significantly reduces redundant search actions and manages massive visual context more effectively.

### C.2. Experimental Settings of Memory Modalities

This part investigates the semantic alignment between observations and memory modalities to resolve the trade-off between compression ratio and critical information preservation. To systematically evaluate how different modalities affect agentic verification, we conduct experiments using four distinct cross-modality strategies within the memory paradigm: **(1) Pre-Captioning** represents a text-only baseline where the entire visual component of the corpus is converted into textual descriptions prior to retrieval. The agent performs reasoning based solely on these pre-generated captions, minimizing token usage at the cost of losing fine-grained visual features. **(2) Visual Observation as Memory** maintains the highest fidelity by directly storing raw multimodal tokens in the context window. While this preserves all visual details, it significantly decreases the signal-to-noise ratio and often leads to context exhaustion in long-horizon tasks. **(3) Context-Aware Captioning** involves retrieving raw multimodal data but memorizing it as dynamic textual summaries. This approach attempts to capture task-relevant visual information in a compressed textual format, though it often suffers from a representation gap during complex verification tasks. **(4) Semantically-Related Visual Memory** implements a selective retention mechanism. After retrieving multimodal data, the agent evaluates the saliency of visual regions and preserves only relevant vision tokens while discarding noise.

### C.3. Experimental Settings of Credit Assignment in Supervision

We explore the reliability of sparse outcome rewards for credit assignment in multi-step agent trajectories. The primary goal is to determine if trajectory-level rewards accurately reflect the validity of individual retrieval and perception steps. Following the protocol in Mem1 (Zhou et al., 2025), we decompose the agent actions into two disjoint subsets: evidence retrieval steps that capture critical clues, and noise or redundant steps that represent irrelevant actions or repeated retrieval using the same query. Specifically, we re-collect all observations throughout the entire reasoning process and employ a direct inference method to evaluate the specific contribution of each observation to the final outcome. To minimize the confounding effects of context length variations on model performance, we utilize Qwen3-VL-Plus as the backbone for this counterfactual evaluation. We conduct the analysis across two dimensions: **(1) For Positive Samples** ($r = 1$)**:** We evaluate trajectories after removing evidence versus removing noise. This comparison is designed to verify the lack of contribution from redundant steps, determining whether the correct final answer persists despite the removal of non-essential actions. **(2) For Negative Samples** ($r = 0$)**:** We test if the model performance recovers by denoising the trajectory to retain only the valid evidence set. This process aims to validate the intrinsic value of these evidence steps, identifying if the initial failure was due to reasoning over accumulated noise rather than a lack of critical information.

## D. Compared Baselines

Here we detailedly introduce the baselines we compare with and our re-produce details.

1. **Vanilla RAG**. During the retrieval phase, it directly uses the original question to search for relevant text, images and videos, which are then inserted into the context to answer the question. Please refer to Appendix J.3 for the detailed prompt.

2. **ReAct RAG** (Yao et al., 2022). The method prompts the RAG agent using a Thought-Action-Observation loop format. Please refer to Appendix J.4 for the detailed prompt.

3. **VideoRAG** (Jeong et al., 2025). This method performs frame selection to extract the information required for inference. We use GVE (Guo et al., 2025) to compute similarity between frames and the query. Although this method is designed for video, embedding model allows us to apply the same coarse-to-fine granularity strategy to both text and images, serving as a reference for performance.

4. **UniversalRAG** (Yeo et al., 2025). It introduces RAG within cross-modal corpora by formulating the task as a routing problem. We use Qwen3VL-8B (4B) as the router to align different settings, and the prompts are borrowed from the original code to ensure a fair comparison.

5. **MemAgent** (Yu et al., 2025a). We implement this method by sequentially feeding the long-context search results into the model's context. Specifically, we directly use the original question to retrieve relevant text, images, and videos, treat the retrieved results understanding as a long-context multimodal understanding task, and then use MemAgent to process this extended context, enabling the model to operate over a broader effective context range than vanilla RAG.

6. **Mem1** (Zhou et al., 2025). This approach updates its memory through a cyclical retrieval-then-memorization process. It is a context-management paradigm that is naturally well-suited for RAG tasks. This method is highly similar to our pilot study in Section 2.2 and follows an iterative summarization paradigm. An approximate version of this effect can be achieved by referring to Appendix J.5. We use the original Mem1 prompt to reproduce the method.

## E. Benchmark Information

We evaluate our method on a comprehensive set of benchmarks covering diverse tasks:

1. **HotpotQA** (Yang et al., 2018) is a large-scale dataset focused on multi-hop question answering that requires reasoning across multiple documents. It contains approximately 113,000 Wikipedia-based question-answer pairs. Unlike datasets constrained by pre-existing knowledge bases, it features diverse natural language questions and provides sentence-level supporting facts to enable systems to provide explainable predictions. The dataset also introduces comparison questions that require models to compare properties of two entities to infer the answer.

2. **SQuAD** (Rajpurkar et al., 2016) is a large-scale reading comprehension dataset consisting of over 100,000 questions created by crowdworkers on a set of Wikipedia articles. Unlike previous datasets that relied on multiple-choice answers or cloze-style tasks, SQuAD requires the model to select a specific segment of text (span) from the reading passage as the answer. This dataset provides a diverse range of answer types, including dates, entities, and clauses, and challenges models to handle significant syntactic differences between the question and the corresponding text in the passage.

3. **WebQA** (Chang et al., 2022) is a multimodal dataset designed to mimic open-domain web search scenarios. It consists of questions that require multi-hop reasoning over both text snippets and images to find the correct answer. Unlike standard VQA tasks where the image is the primary context, WebQA treats images and text as valid knowledge sources that need to be retrieved and combined.

4. **SlideVQA** (Tanaka et al., 2023) is a dataset for document visual question answering focused on understanding slides. It contains over 2,600 slide decks with more than 52,000 slide images and 14,500 questions that require complex reasoning skills such as single-hop, multi-hop, and numerical reasoning. The dataset is designed to support various reasoning types and includes annotated arithmetic expressions for numerical questions to enhance reasoning capabilities.

5. **MMLongbench** (Ma et al., 2024) is a dataset designed to evaluate the document understanding capabilities of VLMs with an emphasis on long-context, multi-modal documents composed of text, images, charts, tables, and layout structures.

6. **LVBench** (Wang et al., 2025c) is a benchmark specifically designed to evaluate long video understanding capabilities. Unlike datasets focused on short clips, it comprises 103 publicly sourced videos with an average duration of approximately 68 minutes, covering diverse categories such as movies, documentaries, and sports. The dataset contains 1,549 manually annotated question-answer pairs that test six core capabilities, including temporal grounding, reasoning, and entity recognition. It is constructed to challenge multimodal models to demonstrate long-term memory and complex reasoning skills required for comprehending extended temporal contexts.

7. **WikiHowQA with HowTo100M** (Bolotova-Baranova et al., 2023; Miech et al., 2019; Jeong et al., 2025) is a composite benchmark constructed to evaluate video-based retrieval and generation tasks. It combines high-quality, human-written instructional questions and answers from the WikiHowQA dataset with the HowTo100M corpus, which consists of millions of instructional videos from YouTube. By associating textual queries with relevant videos, this dataset assesses a system's ability to search for the correct video in a large corpus and generate accurate, visually grounded responses to user questions.

8. **Synthetic QA with HowTo100M** (Jeong et al., 2025; Miech et al., 2019) is a dataset automatically generated to address the lack of training data containing query-video-answer triples for RAG systems. Built upon the HowTo100M corpus, it uses advanced Large Vision-Language Models to create diverse question-answer pairs grounded in specific videos. The questions are designed to be general enough for retrieval tasks—avoiding overly specific frame-level details, while still requiring an understanding of the video content to answer, enabling a comprehensive evaluation of both the retrieval and generation components.

9. **XVBench** is a benchmark designed to address the lack of evaluation standards for cross-video understanding. We construct this dataset using a comprehensive pipeline in Figure 7 that performs fine-grained video segmentation, detailed captioning, and reasoning-graph construction powered by Qwen3-Max. To ensure the quality and appropriate difficulty of the benchmark, we employ embedding distances to rank and filter the samples effectively.

## F. Hyperparameters

The detailed hyperparameters we use during training are shown in Table 5 and Table 6. The construction of the QA data and trajectories is shown in Appendix A. We adopt $\lambda = 0.1$ and $\gamma = 0.3$ for energy dynamics, while the total resource budget is defined as $S_{\text{total}} = 5 \times 256 \times 32 \times 32$ to accommodate high-resolution feature retention. During SFT and RL training, we average the pixels in the multimodal memory bank, and dynamic allocation is enabled only during inference.

*Table 5.* Key hyperparameters for SFT.

| Name | Value |
|---|---|
| Finetuning type | LoRA |
| LoRA Rank | 32 |
| Freeze vision tower | True |
| Freeze multi-modal projector | True |
| Freeze language model | False |
| Cutoff len | 16384 |
| Epochs | 3 |
| Batch size | 8 |
| Gradient accumulation steps | 8 |
| Learning rate | 1.0e-4 |
| LR scheduler type | cosine |

*Table 6.* Key hyperparameters for RL.

| Name | Value |
|---|---|
| Number of agent groups | 8 |
| Batch Size | 32 |
| Mini batch size | 32 |
| Loss Mode | GSPO |
| Learning rate (Actor) | 1.0e-6 |
| KL loss coefficient | 0.001 (optional) |
| Tensor model parallel size | 4 |
| Total epochs | 2 |
| Max prompt length | 20240 |
| Max response length | 512 |
| GPU memory utilization | 0.6 |

# G. Case Study

In Figure 8 and Figure 9, we describe the reasoning paths of VimRAG to show how our framework builds and updates the Multimodal Memory Graph. These cases reveal two main challenges in long multimodal retrieval tasks. The first challenge is solving the state blindness problem to search large video datasets effectively without processing irrelevant data. The second challenge involves the dynamic resolution allocation mechanism that keeps detailed visual information while meeting token limits.

# H. Limitations

Despite our best efforts, this paper still has some limitations. First, enhancing the capabilities of the base model can contribute to the overall improvement of the system. Second, the current multi-turn interactions may not meet the requirements of high real-time applications. Finally, the accuracy of the current retriever still needs improvement to better support the RAG system.

# I. Ethics Statement

This work focuses on improving the efficiency and accuracy of multimodal agentic systems. All datasets used in our experiments, such as HowTo100M and HotpotQA, are publicly available and do not contain private or sensitive personal information. Furthermore, by optimizing token allocation and reducing unnecessary computation, our approach contributes to more energy-efficient AI systems, aligning with the goals of sustainable computing.

# J. Prompts

In this section, we illustrate all the prompts used in our paper.

### J.1. Model-based Evaluation Prompt

See Figure 12.

### J.2. Question Verifier Prompt

See Fogure 16

### J.3. Vanilla RAG Prompt

See Figure 11.

### J.4. ReAct Prompt

See Figure 10.

### J.5. Summarization as Memory Prompt

See Figure 13. The memory is directly updated by the summarized information.

### J.6. Graph as Memory Prompt

See Figure 14. The memory contains semantic graph only.

### J.7. VimRAG Prompt

See Figure 15.

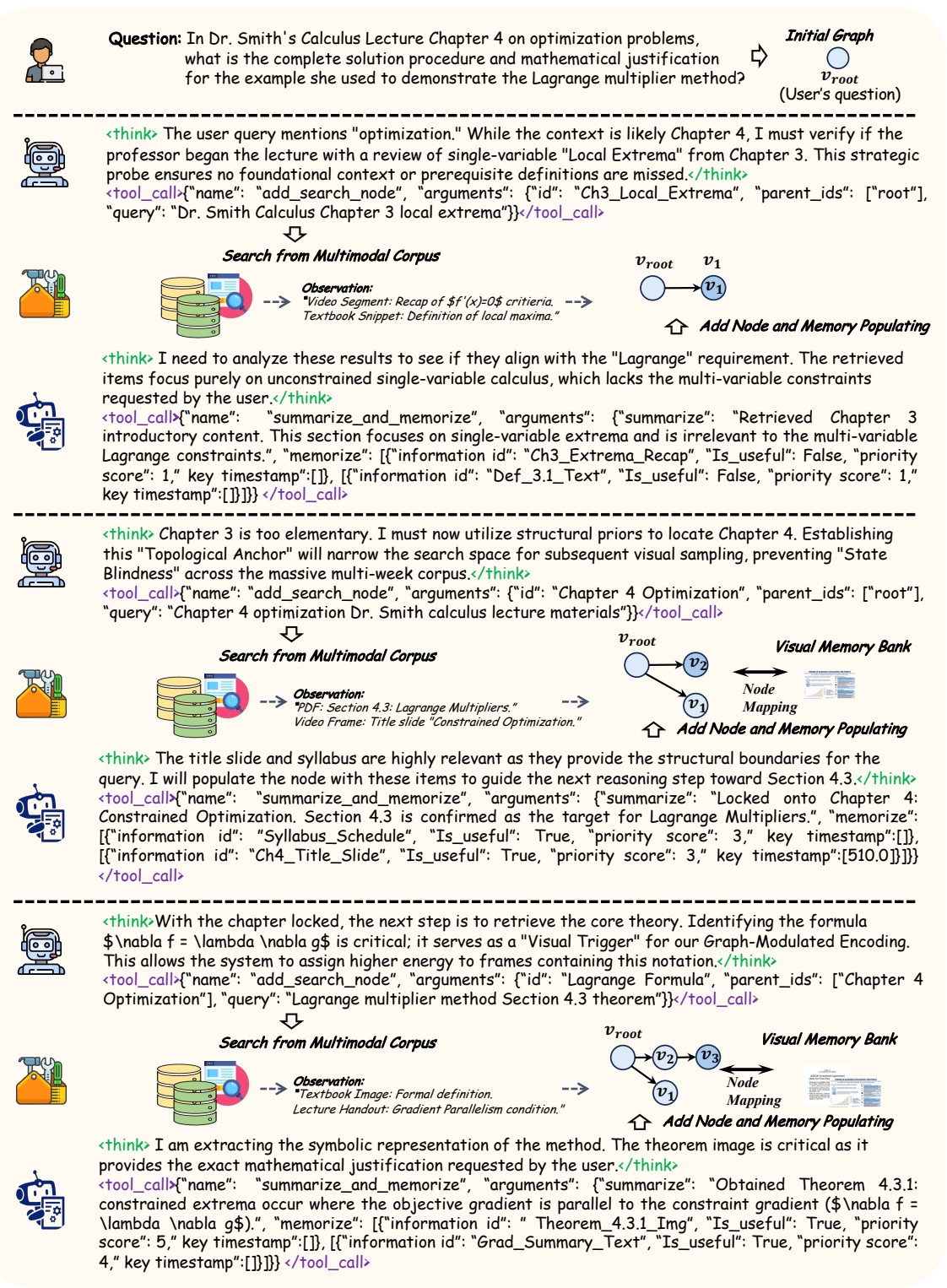

*Figure 8.* **Case Study (Part I).** The agent initializes the Multimodal Memory Graph to address a complex query regarding a calculus lecture.

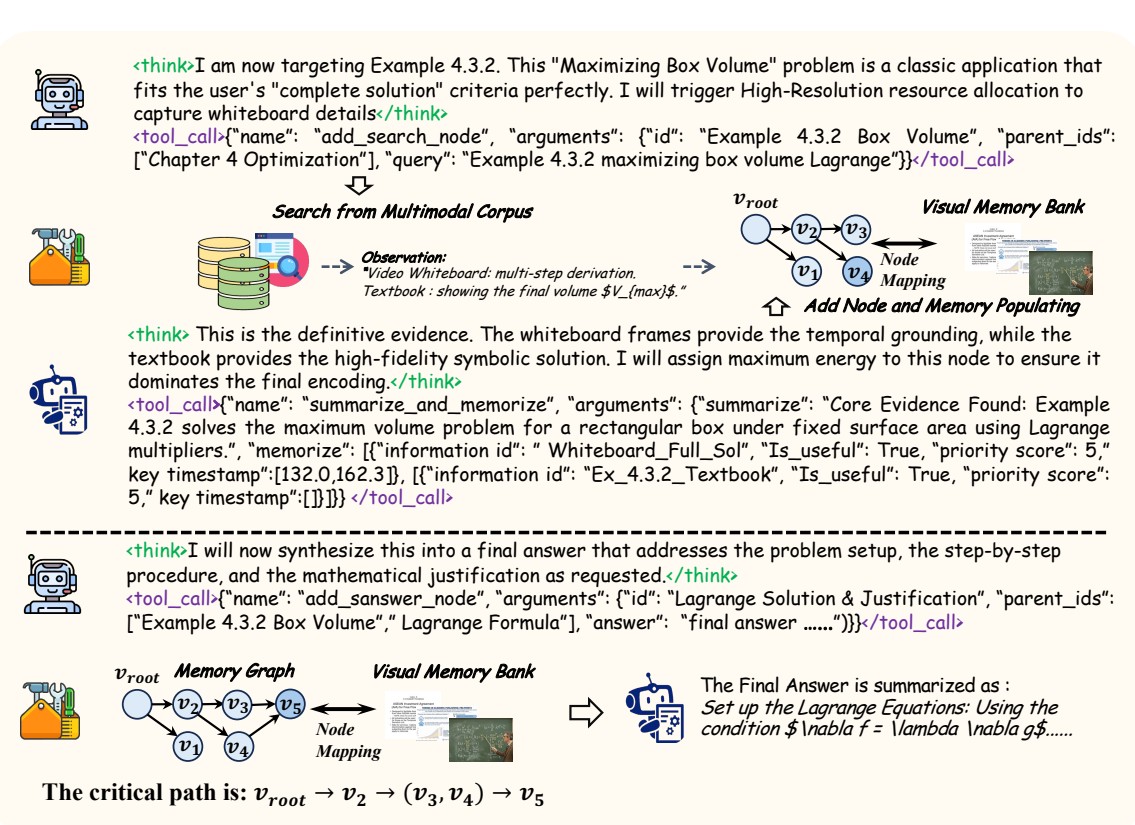

<think>I am now targeting Example 4.3.2. This "Maximizing Box Volume" problem is a classic application that fits the user's "complete solution" criteria perfectly. I will trigger High-Resolution resource allocation to capture whiteboard details</think>
<tool_call>{"name": "add_search_node", "arguments": {"id": "Example 4.3.2 Box Volume", "parent_ids": ["Chapter 4 Optimization"], "query": "Example 4.3.2 maximizing box volume Lagrange"}}</tool_call>

**Search from Multimodal Corpus**

**Observation:**
"Video Whiteboard: multi-step derivation.
Textbook : showing the final volume $V_{max}$."

$v_{root}$ **Visual Memory Bank**

*Node Mapping*

**Add Node and Memory Populating**

<think> This is the definitive evidence. The whiteboard frames provide the temporal grounding, while the textbook provides the high-fidelity symbolic solution. I will assign maximum energy to this node to ensure it dominates the final encoding.</think>
<tool_call>{"name": "summarize_and_memorize", "arguments": {"summarize": "Core Evidence Found: Example 4.3.2 solves the maximum volume problem for a rectangular box under fixed surface area using Lagrange multipliers.", "memorize": [{"information id": " Whiteboard_Full_Sol", "Is_useful": True, "priority score": 5," key timestamp":[132.0,162.3]}, [{"information id": "Ex_4.3.2_Textbook", "Is_useful": True, "priority score": 5," key timestamp":[]}]}} </tool_call>

----

<think>I will now synthesize this into a final answer that addresses the problem setup, the step-by-step procedure, and the mathematical justification as requested.</think>
<tool_call>{"name": "add_sanswer_node", "arguments": {"id": "Lagrange Solution & Justification", "parent_ids": ["Example 4.3.2 Box Volume"," Lagrange Formula"], "answer": "final answer ......"}}</tool_call>

$v_{root}$ **Memory Graph** **Visual Memory Bank**

*Node Mapping*

The Final Answer is summarized as :
*Set up the Lagrange Equations: Using the condition $\nabla f = \lambda \nabla g$......*

**The critical path is: $v_{root} \rightarrow v_2 \rightarrow (v_3, v_4) \rightarrow v_5$**

*Figure 9.* **Case Study (Part II).** The final answer is synthesized by traversing the critical path ($v_{root} \rightarrow v_2 \rightarrow (v_3, v_4) \rightarrow v_5$).

---

**ReAct Prompt**

## System Prompt:

You are an expert assistant who can solve any question using tool calls. To do so, you have been given access to search engines and tools.

**Available Tools**

1. **search:** Collect relevant information based on the query. (Params: keywords; Returns: search results).
2. **answer:** Respond directly to the user based on search results. (Params: response).

**Workflow**

1. Comprehend the user's query and identify the core points of inquiry.
2. Formulate clear and specific search strings using the `search` function.
3. Compose a clear and concise final response based on the information obtained.

**Strictly Prohibited Behaviors**

1. Do not provide answers using information not obtained through the designated tools.
2. Do not fabricate or extrapolate beyond the content returned by the tools.
3. Do not output vague summaries or unverified speculations.
4. Do not call the search engine and give the answer in the same response.

**Reply Format**

You **must** respond with the following format:
*Option 1: Searching*
`<thinking>`Your reasoning process`</thinking>`
`<search>`Your search query`</search>`
*Option 2: Answering*
`<thinking>`Your reasoning process`</thinking>`
`<answer>`Your detailed response`</answer>`

- - - - - - - - - - - - - - - - - - - - - - - - - - - - - - - - - - - - - - - - - - - -

## User Prompt:

**Execution Instructions**

1. You must conduct reasoning inside `<thinking>` tags before answering or searching.
2. If you lack knowledge, call the search engine using `<search>` tags.
3. You may search as many times as needed.
4. Once sufficient information is gathered, provide the final answer inside `<answer>` tags.

**Required Response Format**
*When searching:*
`<thinking>`Your reasoning process`</thinking>`
`<search>`Your search query`</search>`
*When answering:*
`<thinking>`Your reasoning process`</thinking>`
`<answer>`Your detailed response`</answer>`

**User Query**
{Query Description}

*Figure 10.* Prompt of ReAct.

---

**Vanilla RAG Prompt**

**System Prompt:**
Answer the given question. You must conduct reasoning inside `<thinking>` and `</thinking>` first every time you get new information. After reasoning, you should directly provide the answer inside `<answer>` and `</answer>`, without detailed illustrations. For example, `<answer>` Beijing `</answer>`.

- - - - - - - - - - - - - - - - - - - - - - - - - - - - - - - - - - - - - - - - - - - - - - - - - - - - - -

**User Prompt:**
**Query**
{Query Description}
**Retrieved Multimodal Information**
{Retrieved Videos / Images / Text Tokens}

*Figure 11.* Prompt of Vanilla RAG.

---

**Reward Model Prompt**

**System Prompt:**
**Character Introduction**
You are an expert evaluation system for a question answering chatbot.
You are given the following information:
- the query
- a generated answer
- a reference answer
Your task is to evaluate the correctness of the generated answer.
**Response Format**
Your response should be formatted as following: `<judge>`True or False`</judge>`
If the generated answer is correct, please set "judge" to True. Otherwise, please set "judge" to False.
Please note that the generated answer may contain additional information beyond the reference answer.

- - - - - - - - - - - - - - - - - - - - - - - - - - - - - - - - - - - - - - - - - - - - - - - - - - - - - -

**User Prompt:**
Query: {Query Description}
Reference Answer: {Reference Answer}
Generated Answer: {Generated Answer}

*Figure 12.* Prompt of Model-based Reward.

Iterative Summarization as Memory Prompt

**System Prompt:**
You are an expert assistant who can solve any question using tool calls. You are presented with a problem and a previous memory.
To do so, you have been given access to search engine.
### Available Tools
search:
- Collect relevant information based on the query.
- Parameters: The keywords or question to search for.
- Returns: Search results for query.
### Workflow
1. Comprehend the user's query and identify the core points of inquiry.
2. Formulate clear and specific search strings to retrieve relevant information using the `search` function.
3. Every time you call the search engine, you need to update the memory according to the search results and the current memory.
4. Compose a clear and concise final response if the information is sufficient.
### Requirements
1. Ensure tool usage is precise and queries are well-formulated.
2. Provide accurate and well-structured answers to user queries.
3. Iterate search attempts if initial results are insufficient.
4. You can only provide a final answer or use a search engine, but not both in the same response.
5. You must call the search engine to get the search results at least once.
6. Follow the response format.
### Strictly Prohibited Behaviors:
1. Providing answers using information not obtained through the designated tools.
2. Fabricating or extrapolating beyond the content returned by the tools.
3. Outputting vague summaries, hypothetical judgments, or unverified speculations.
4. Repeatedly using semantically similar queries when calling the search engine.
5. Do not call the search engine and give the answer in the same response.
### Reply Format
You **must** response with the following format:
When you need to search, you need to provide the search query in the following format:
`<think>Your reasoning process</think>`
`<search>Your search query</search>`
When you need update memory, you need to provide the summary in the following format:
`<update_memory>Updated memory</update_memory>`
When you have gathered enough information to answer the question, provide your response immediately:
`<think>Your reasoning process</think>`
`<answer>Your detailed response to the question</answer>`

- - - - - - - - - - - - - - - - - - - - - - - - - - - - - - - - - - - - - - - - - - -

**User Prompt:**
Question: {query}
Memory: {memory}

- - - - - - - - - - - - - - - - - - - - - - - - - - - - - - - - - - - - - - - - - - -

**Memory Update Prompt:**
Please give the updated memory immediately: `<update_memory>Updated memory</update_memory>`
**Retrieved Multimodal Information**
{Retrieved Videos / Images / Text Tokens}

*Figure 13.* Prompt of Iterative Summarization as Memory.

Graph as Memory Prompt

## System Prompt:

You are an intelligent agent designed to solve user queries by either answering directly or iteratively searching for information. Your goal is to build a Directed Acyclic Graph (DAG) that represents the search process for a single user query, where each node corresponds to a step in reasoning or information gathering.

**Graph Nodes**

The graph has three types of nodes:
• `root`: The initial node representing the user's original question.
• `search`: A node representing a search query issued to an external search engine. Each search node must have a unique, highly summarized title that captures the intent of the query and must be significantly different from previous queries.
• `answer`: The final node where you provide a complete answer to the user's question. This node does not have an ID.

**Rules**

1. You can only add one node per turn.
2. Each `search` node must: (a) Have a unique `id` (title) that is a short, descriptive phrase summarizing the query intent; (b) Be connected to its parent via a directed edge (specify `parent_id`); (c) Contain a `query` field with the actual search string; (d) The query must be substantially different from prior ones.
3. After issuing a search query, you will receive results. Then, you must summarize the relevant content from those results into a concise `summary` (which will be added externally to the node).
4. You must decide at each step whether to: Answer directly (output an `answer` node), OR Search (output a `search` node with a new query).
5. Queries must be substantially different from prior ones—avoid redundancy or rephrasing the same idea.
6. When generating a `search` node, use the `add_search_node` function.
7. When receiving search results, you can summarize the results with the `summary_search_node` function.
8. Once you believe you have enough information to answer the question, please output an `add_answer_node` function call.

**Available Tools**

1. **add_search_node**
**Description:** Creates a new search node in the graph. This tool should be used to issue a search query to an external engine. Each node must have a unique, summarized ID reflecting its intent.
**Parameters:**
- `id`: A unique, short, and descriptive title for the node capturing the intent of the query.
- `parent_ids`: The ID(s) of the previous node(s) this search stems from.
- `query`: The actual search query to be executed. Must be substantially different from all prior queries.

2. **add_answer_node**
**Description:** Creates the final node in the graph containing the complete and final answer to the user's original question.
**Parameters:**
- `parent_ids`: The ID(s) of the node(s) that provided the information necessary for this final answer.
- `answer`: The comprehensive and complete final answer to the user's question.

3. **summarize_and_memorize**
**Description:** Mandatory tool that MUST be called after every 'add_search_node' without exception. It acts as the finalization step for a search node, filtering raw data into high-density memory.
**Parameters:**
- `summarize`: A 1-3 sentence synthesis focusing strictly on facts that directly address the user's intent. If the search returned no relevant information, explicitly state so.

**Process Flow**
- Start with the user's query as the root node.
- On each turn, evaluate whether you can answer now or need more info.
- If searching: emit a `add_node` MCP command with a new `search` node.
- After search returns: emit a `summary` MCP command with insights from the result.
- Repeat until you can answer.

**Important:** Only one action per turn. Do not combine actions. Always follow the MCP format strictly.

**Reply Format**
For each function call, return a json object with function name and arguments within `<tool_call></tool_call>` XML tags and your reasoning process in `<thinking></thinking>` XML tags:

```
<thinking>
Your reasoning process ...
</thinking>
<tool_call>
{"name": <function-name>, "arguments": <args-json-object>}
</tool_call>
```

**User Prompt:**

**### User Query**
{Query Description}
**### Agent Action Graph**
{Semantic Action Graph}

**Memorize Prompt:**
Now that the search results for the query have been returned, please analyze them carefully and provide a concise, factual summary of the information that is directly relevant to answering the original user question or give the answer node if the information is sufficient to answer the question.
**Your summary should:**
- Be brief (1-3 sentences).
- Focus only on key insights or facts from the results.
- Avoid redundancy or speculation.
- Highlight how this information contributes to solving the problem or narrowing down the answer.
- Not repeat what was already known or covered in prior nodes.
**Retrieved Multimodal Information**
{Retrieved Videos / Images / Text Tokens}

*Figure 14.* Prompt of Graph as Memory.

VimRAG Prompt

## System Prompt:

You are an intelligent agent designed to solve user queries by either answering directly or iteratively searching for information. Your goal is to build a Directed Acyclic Graph (DAG) that represents the search process for a single user query, where each node corresponds to a step in reasoning or information gathering.

### Graph Nodes

The graph has three types of nodes:
• `root`: The initial node representing the user's original question.
• `search`: A node representing a search query issued to an external search engine. Each search node must have a unique, highly summarized title that captures the intent of the query and must be significantly different from previous queries.
• `answer`: The final node where you provide a complete answer to the user's question. This node does not have an ID.

### Rules

1. You can only add one node per turn.
2. Each `search` node must: (a) Have a unique `id` (title) that is a short, descriptive phrase summarizing the query intent; (b) Be connected to its parent via a directed edge (specify `parent_id`); (c) Contain a `query` field with the actual search string; (d) The query must be substantially different from prior ones.
3. After issuing a search query, you will receive results. Then, you must summarize the relevant content from those results into a concise `summary` (which will be added externally to the node).
4. You must decide at each step whether to: Answer directly (output an `answer` node), OR Search (output a `search` node with a new query).
5. Queries must be substantially different from prior ones—avoid redundancy or rephrasing the same idea.
6. When generating a `search` node, use the `add_search_node` function.
7. When receiving search results, you can summarize the results with the `summary_search_node` function.
8. Once you believe you have enough information to answer the question, please output an `add_answer_node` function call.

### Available Tools

1. **add_search_node**
**Description:** Creates a new search node in the graph. This tool should be used to issue a search query to an external engine. Each node must have a unique, summarized ID reflecting its intent.
**Parameters:**
   - `id`: A unique, short, and descriptive title for the node capturing the intent of the query.

   - `parent_ids`: The ID(s) of the previous node(s) this search stems from.

   - `query`: The actual search query to be executed. Must be substantially different from all prior queries.

2. **add_answer_node**
**Description:** Creates the final node in the graph containing the complete and final answer to the user's original question.
**Parameters:**

   - `parent_ids`: The ID(s) of the node(s) that provided the information necessary for this final answer.

   - `answer`: The comprehensive and complete final answer to the user's question.

3. **summarize_and_memorize**
**Description:** Mandatory tool that MUST be called after every 'add_search_node' without exception. It acts as the finalization step for a search node, filtering raw data into high-density memory. This tool must be executed even if the retrieved information is entirely irrelevant to the user's query to formally close the current search cycle.
**Parameters:**

- `summarize`: A 1-3 sentence synthesis focusing strictly on facts that directly address the user's intent. If the search returned no relevant information, explicitly state so.

- `memorize`: An exhaustive list of items (Text, Image, Video). Each item includes:

  - `information_id`: Unique identifier (e.g., 'Text 1').
  - `is_useful`: Boolean judgment of value.
  - `key_timestamp`: Array of seconds (for Video) or empty.
  - `priority_score`: 1 (marginal) to 5 (critical).

**Process Flow**
- Start with the user's query as the root node.
- On each turn, evaluate whether you can answer now or need more info.
- If searching: emit a `add_node` MCP command with a new `search` node.
- After search returns: emit a `summary` MCP command with insights from the result.
- Repeat until you can answer.

**Important:** Only one action per turn. Do not combine actions. Always follow the MCP format strictly.

**Reply Format**
For each function call, return a json object with function name and arguments within `<tool_call></tool_call>` XML tags and your reasoning process in `<thinking></thinking>` XML tags:

```
<thinking>
Your reasoning process ...
</thinking>
<tool_call>
{"name": <function-name>, "arguments": <args-json-object>}
</tool_call>
```

---

## User Prompt:

**### User Query**
{Query Description}
**### Agent Action Graph**
{Semantic Action Graph}
**### Multimodal Memory Bank**
{Vision Tokens}

---

## Memorize Prompt:

Now that the search results for the query have been returned, please analyze them carefully and provide a concise, factual summary of the information that is directly relevant to answering the original user question or give the answer node if the information is sufficient to answer the question.

**Your summary should:**
- Be brief (1-3 sentences).
- Focus only on key insights or facts from the results.
- Avoid redundancy or speculation.
- Highlight how this information contributes to solving the problem or narrowing down the answer.
- Not repeat what was already known or covered in prior nodes.

**Retrieved Multimodal Information**
{Retrieved Videos / Images / Text Tokens}

*Figure 15.* Prompt of VimRAG. The model performs retrieval or generates an answer based on the System Prompt and User Prompt. If retrieval is triggered, it adds nodes to the graph under the guidance of the Memorize Prompt.

---

Question Verifier Prompt

**System Prompt:**
I have some QA data here, and you can observe that the questions can be divided into two categories:

**The category #A:** When you see this question alone without a given document, you are sure to find a unique document in a corpus to provide a unique answer. The question having some key words to help you locate the document from corpus.

**The category #B:** When you see this question alone without a given document, you will find hard to locate a document to give a deterministic answer for this question, because you will find multiple candidate documents in a corpus, which may lead to different answers for this question. The question do not have any special key words to help you locate the document from corpus.

**Examples:**
The number mentioned on the right of the leftside margin? #B
What is the date mentioned in the second table? #B
What is the full form of PUF? #A
What is the number at the bottom of the page, in bold? #B
Who presented the results on cabin air quality study in commercial aircraft? #A
What is the name of the corporation? #B
To whom this is addressed? #B
What is the source? #B
What is the heading of the document? #B
What is the subject? #B
what mm Marlboro Menthol were subjectively smoked by the Richmond Panel? #A
What sort of communication/letter is this? #B
According to the listed requirements, what must be the age group of female smokers? #A
During the process of prototype production and ringtipping, some cigarettes were observed to have burn holed in which paper? #A
How many distinct mechanisms appear to play a role in the breakup of a smoke column into a multi-dimensional flowfield? #A
Where was the conference held? #B
Who is in cc in this letter? #B
Under BOLD, primary production of Blend #24- will be completed by which date? #A
What are the steps of Weft Preparation between Spinning bobbin and Weaving? #A
What are the three types of Fabric manufacturing technology? #A
What level comes between Middle Managers and Non-managerial Employees? #A
What are the six parts of COLLABORATION MODEL of the organization where James has a role of leading the UK digital strategy? #A
What are the six parts of CONCERN'S COLLABORATION MODEL? #A
Transparent process of Land Acquisitions is an example of what? #A
What is Old Act Section 4 in the New Act Section? #A
Do updates or the activities that are shorter and more intense than projects, and are frequently supported with offline media have lower intensity? #A
Is differentiation high or low in a market in which not many people are interested in this content, but it does differentiate you in the market? #A

- - - - - - - - - - - - - - - - - - - - - - - - - - - - - - - - - - - - - - - - - - - - - - - - - - - -

**User Prompt:**
Label the following question as either #A or #B based on these criteria:
Here is the question: {question}
You should put the category of the question following the format: #A or #B.

*Figure 16.* Prompt for Question Verifier.

