# OpenReview forum: "Navigating Massive Visual Context in Retrieval-Augmented Generation via Multimodal Memory Graph"
_ICML.cc/2026/Conference — ICML 2026 regular_

### Official Review · Reviewer_VnSk · 2026-02-19

**Soundness:** 3
**Presentation:** 3
**Significance:** 3
**Originality:** 3
**Overall Recommendation:** 5
**Confidence:** 3

**Summary:**

The paper proposes VimRAG, a multimodal Retrieval-Augmented Generation framework that models an agent’s iterative reasoning as a dynamic directed acyclic “Multimodal Memory Graph.” On top of this structure, it introduces Graph‑Modulated Visual Memory Encoding to allocate vision token budgets according to a node’s topological salience and a recursive energy formulation, and Graph‑Guided Policy Optimization that prunes steps for fine‑grained credit assignment during RL.

**Compliance With Llm Reviewing Policy:**

Affirmed.

**Final Justification:**

The paper presents a well-motivated graph-based multimodal RAG framework with thoughtful designs for token allocation and credit assignment. My main concerns about backbone dependence and evaluation robustness are largely addressed in the rebuttal through additional experiments across architectures and clearer metric justification. Overall, the work is technically sound and reasonably general; the rebuttal strengthens confidence but does not change my original assessment, and I maintain my score.

**Key Questions For Authors:**

1. Why was Qwen3VL-30B-A3B-Instruct chosen as the backbone model? Does this choice suggest that MoE architectures are more suitable than dense instruction-tuned models for this task?

**Limitations:**

1. Evaluation relies on a model-based binary judge (Qwen3-Max) across tasks including those with exact-match metrics, introducing potential bias and hindering comparability with prior work.

**Strengths And Weaknesses:**

1. Modeling the agent’s state as a multimodal, evolving DAG is a thoughtful extension of graph-based reasoning to RAG in genuinely multimodal settings.

2. The energy-based visual token allocation tied to graph topology and temporal decay is an interesting and plausible way to reconcile token budgets with visual evidence fidelity.

3. Graph-guided pruning for step-wise credit assignment is a clean mechanism to reduce gradient noise from irrelevant steps in long trajectories.

---

> ### Author Rebuttal · Authors · 2026-03-31
>
> Dear Reviewer VnSk,
>
> We sincerely appreciate your insightful review and the valuable insights you provided. Your recognition of our work's strengths has been a great encouragement to our team. We have tried our best to address all concerns in the past few days. Please see our responses to each point below:
>
> >**Question: About the selection of base model (MoE vs. Dense).**
>
> Many thanks for your highly insightful suggestions! In our agent workflow, the construction of the Multimodal Memory Graph and the Graph-Guided Policy Optimization are fundamentally decoupled from the model architecture. These components operate at the semantic and logical levels of the reasoning process rather than relying on any specific parameter activation patterns. Therefore, whether the backbone is a dense model or an MoE model, VimRAG can be effectively adapted and can fully leverage its strengths.
>
> To systematically analyze and address your questions regarding the architectural choices, we have supplemented our work with experiments across different model architectures. The additional results on various architectures further confirm the generalizability of our method, as detailed below:
>
> | Model | Method | Text | Image | Video | Overall |
> | --- | --- | --- | --- | --- | --- |
> | Qwen3-VL-4B | ReAct | 63.4  | 32.7  | 19.4  | 33.6  |
> |  | Mem1 | 69.2  | 40.3  | 26.5  | 40.6  |
> |  | VimRAG | 74.4  | 42.8  | 32.5  | 45.2  |
> | Qwen3-VL-8B | ReAct | 68.2  | 35.1  | 24.5  | 37.7  |
> |  | Mem1 | 70.7  | 44.3  | 29.5  | 43.6  |
> |  | VimRAG | 77.8  | 49.9  | 36.5  | 50.1  |
> | Qwen3-VL-30B-A3B | ReAct | 71.0  | 38.7  | 26.9  | 40.6  |
> |  | Mem1 | 72.6  | 46.9  | 31.0  | 45.5  |
> |  | VimRAG | 79.2  | 50.7  | 40.3  | 52.4  |
>
> Experimental results show that VimRAG delivers consistent performance improvements on both MoE and dense architectures, demonstrating strong robustness across diverse model designs. This flexibility allows our framework to be deployed seamlessly, from edge devices running small dense models to high-throughput services powered by large MoE models, without requiring architecture-specific modifications.
>
> >**Limitation: About the robustness of a model‑based evaluator and the consistency of evaluations across different models.**
>
> Thank you for your highly valuable questions, which have been instrumental in enhancing our work. We used different metrics to evaluate the experimental results, including EM, Bert Score and Model-Based Accuracy, and analyzed these metrics.
>
> | Method | EM | Bert Score | Model-Based Evaluation Accuracy |
> | --- | --- | --- | --- |
> | ReAct | 0.9% | 0.452 | 37.7% |
> | Mem1 | 2.8% | 0.477 | 43.6% |
> | VimRAG | 3.3% | 0.538 | 50.1% |
>
> Notably, the exact‑match metric is suitable for mathematical tasks that require precise values, but it fails to evaluate performance effectively in most RAG QA tasks, where answers are typically presented in a sentence and may be expressed in different valid ways. It is widely recognized that using advanced LLMs as evaluators provides a fairer and more accurate assessment.

---

> > ### Author Rebuttal · Reviewer_VnSk · 2026-03-31
> >
> > Thank you for the detailed rebuttal. The additional experiments across dense and MoE models help support the generality of the method, and the inclusion of multiple evaluation metrics is appreciated. Overall, the rebuttal is helpful, and I will maintain my score.

---

> > > ### Author Response · Authors · 2026-04-01
> > >
> > > Dear Reviewer VnSk:
> > >
> > > We appreciate your time and effort in reviewing our manuscript. We are very grateful for your recognition and support of our work. Your review has been of great help to us in improving our manuscript.
> > >
> > > **Many thanks for your constructive comments, time, and patience.**
> > >
> > > Best regards and thanks,
> > >
> > > The Authors

---

### Official Review · Reviewer_DeVE · 2026-02-21

**Soundness:** 3
**Presentation:** 2
**Significance:** 3
**Originality:** 3
**Overall Recommendation:** 4
**Confidence:** 2

**Summary:**

The paper VimRAG proposes a new paradigm for RAG in large-size corpus/context. The core idea is a graph-like structure that is quite similar to how human search for information: we may search into dead end, but we go back and search again for other directions. To process long and low density visual tokens, the paper proposes to dynamically allocate token budgets to selective visual tokens. With an intuitive method to allocate token budges using energy computations, VimRAG give different budgets to different nodes on the graph: dead-end/old/less relevant nodes receives less budgets and vice versa. This graph structure is also used during training to punish dead-end behaviors. Experimentally, this graph-based RAG proves better than linear methods like ReAct or other compression methods.

**Compliance With Llm Reviewing Policy:**

Affirmed.

**Final Justification:**

The authors resolves my questions. But since I'm not an expert in this field, I'll just keep weak accept with low confidence.

**Key Questions For Authors:**

1, In Section 3.1 Exploratory Expansion (line 223-226), how is the retrieval of O_t from q_t actually achieved? From external search engine provided by the benchmark? What is the general size of O_t?

2, Section 3.2 Recrusize Reinforcement. Here is contradictory for me. In intrinsic energy, you add a temporal decay to mimic the behavior of human forgetting, that gives lower weight to older nodes. But here you add the sum of the child nodes back to the parent node. In most cases, older nodes will have more child nodes, so this contradicts with the intuition of temporal decay. Could the authors explain this design choise and the intuitions?

3, Lack of more qualitative results and comparisons. The only qualitative result is in Appendix G. But in that example, there is only one dead end (v1), so in my understanding, giving less tokens to v1 is the only advantage of graph structure than just linear structure. Is this really a significant advantage? In this example, the linear search path would be v_root -> v1 -> v2 ... v5, which is not much heavier than the path in a graph memory. So is there any cases your method can handle while other baselines cannot?

4, In appendix F, you mentioned ``During SFT and RL training, we
average the pixels in the multimodal memory bank, and dynamic allocation is enabled only during inference.'' Could you explain this choice?

5, You have a total token budget S_total, and I believe you keep this value fixed for all the experiments of VimRAG. Then how did you apply this budget to other baselines (to ensure fair comparison)? Say for a baseline method that has content larger than this budget, did you just naturally let it degrade in performance, or you also fix the max token in some way?

**Limitations:**

See above.

**Strengths And Weaknesses:**

Strengths:

The paper is well written. I am not an expert in LLM, but I can get the general idea of this paper. The flow is systemmatic. Especially for section2, it explains the motivations and shows other paradigms clearly with experiments and graphs.

Weakness:

In terms of writing, some ideas in the methodology part is not conveyed clearly. Say in section 3.1, it seems that the nodes are still constructed linearly. Only when I read the appendix that I get why it generates a graph structure.

---

> ### Author Rebuttal · Authors · 2026-03-31
>
> Dear Reviewer DeVE,
>
> We sincerely appreciate your insightful review and the valuable insights you provided. Your recognition of our work's strengths has been a great encouragement to our team. We have tried our best to address all concerns. Please see our responses to each point below:
>
> >**W: About the description of the memory process.**
>
> We are glad to hear that the appendix helped clarify our method. We will further refine the presentation in the revision to convey our insights more clearly and accurately.
>
> >**Q1: More explanation of external search engines and practical implementation.**
>
> The search engine is built on the embedding model described in Appendix B. All baselines and our VimRAG interact with the same retriever to ensure a fair comparison. The retrieval process in two steps:
>
> 1. Build the index offline: All multimodal entities in the corpus are pre‑encoded into vectors using the embedding model.
> 2. Online Retrieval: The query $q_t$ is encoded into vector by the embedding model, and then the similarity is computed. The top‑K entries with the highest similarity are selected as the observations.
>
> Each observation $O_t$ has a different token length depending on its modality and actual content. This setting is consistent with real‑world applications.
>
> >**Q2: More analysis on the Memory Energy Formulation.**
>
> Thank you for this sharp observation. However, this is not a conflict but rather an intentional high‑pass filter design that selectively removes noise while preserving critical evidence.
>
> |Type|Node Characteristics|What Happens|Example|
> |-|-|-|-|
> |Low energy|Old and Isolated (no children)|Energy drops to near-zero → Compressed or dropped|Dead-end steps or irrelevant outdated nodes.|
> |High energy|New observations or have active descendants|High energy → Dense visual tokens|Key definitions or evidence|
>
> Our goal is not to fully simulate human forgetting, but to leverage this paradigm to filter out nodes that are both old and lack child nodes, thereby removing noise such as isolated, outdated, and irrelevant observations.
>
> >**Q3: More qualitative results and comparisons to better highlight the advantages of the graph structure.**
>
> We would like to further clarify that the graph structure offers capabilities beyond the example presented in Appendix G. Its topological flexibility enables a variety of reasoning patterns.
>
> |Scenario/Task|Graph Structure Advantage|Limitations of Baselines|
> |-|-|-|
> |Backtracking|New nodes can attach to any historical node, enabling dynamic path restructuring.|Linear structures can only accumulate sequentially, leading to state blindness and invalid actions (Fig. 2b and Sec. 2.2).|
> |Iterative Refinement in Long Reasoning|Explicit chains via parent pointers, preserving complete historical state traces.|Iterative summarization destroys intermediate reasoning states, leading to the loss of earlier retrieval intentions.|
> |Repeated Verification|Graphs make it easy to trace back past critical evidence.|No chance to selectively reinforce valuable early observations, leading to context exhaustion.|
>
> Crucially, the graph topology allows the agent to track multiple possibilities at once and prune dead paths without losing structural context, and it is a necessary structural prior for long-horizon multimodal reasoning.
>
> >**Q4: The explanation of the settings for multimodal memory allocation during training and inference.**
>
> We consider dynamic token allocation a form of context management. During training, we encourage the model to learn comprehensive representations, while during inference, we selectively expose only the most important features. This approach gives the model strong generalization and perceptual capabilities.
>
> + Training: (i) During SFT, fixed token lengths ensure that the student model learns from consistent vision language representation pairs produced by the teacher model. (ii) During RL rollouts and updates, the training focuses on its agentic reasoning capabilities rather than perception, so this type of context‑management paradigm is not required.
> + Inference: At test time, by allocating high‑resolution tokens to the most critical graph nodes, we obtain the strongest possible reasoning performance within the available context window.
>
> Following this principle, we perform stable, compute‑intensive training and then conduct inference with token allocation to balance the trade‑off between efficiency and performance.
>
> >**Q5: About the experimental details and the baseline setting.**
>
> To ensure a fair comparison, we evaluated each baseline under its own optimal conditions and paradigm without any limits, such as ReAct’s linear history or Mem1’s iterative summarization. The total token budget is specific to the visual memory mechanism, where it serves as a dynamic resource pool for adaptive vision‑token allocation. The results in Tab. 2 and Fig. 6 show that VimRAG achieves better performance with lower latency and improved agent‑specific capabilities.

---

> > ### Author Rebuttal · Reviewer_DeVE · 2026-04-01
> >
> > Thanks for your kind explanation. It answers my questions. Since I'm not an expert in this area, I will just keep my WA score.

---

> > > ### Author Response · Authors · 2026-04-02
> > >
> > > Dear Reviewer DeVE:
> > >
> > > We sincerely appreciate your positive feedback and are very glad to hear that our explanations have fully addressed your questions and concerns.
> > >
> > > We truly value the perspective you brought to our work, which has helped us improve the clarity of our manuscript.
> > >
> > > **Many thanks for your constructive comments, time, and patience.**
> > >
> > > Best regards and thanks,
> > >
> > > The Authors

---

### Official Review · Reviewer_sPDG · 2026-03-04

**Soundness:** 3
**Presentation:** 3
**Significance:** 3
**Originality:** 3
**Overall Recommendation:** 6
**Confidence:** 4

**Summary:**

This paper proposes VimRAG, an agentic memory framework for multimodal Retrieval-Augmented Generation (RAG). Instead of relying on linear interaction histories, the framework organizes the reasoning and retrieval process as a dynamic multimodal memory graph (DAG) in order to better manage large visual contexts.
Specifically, the method reformulates the reasoning process into a dynamic DAG memory structure. Based on the graph topology and semantic relevance scores, the framework computes an energy score for each node and adaptively allocates token density for visual evidence. High-resolution tokens are preserved for critical evidence while low-value information is compressed or discarded.
The paper also introduces a Graph-Guided Policy Optimization mechanism that performs graph pruning and gradient masking by identifying key reasoning paths and valuable retrieval nodes. This design aims to improve credit assignment in multi-step reasoning trajectories and increase training efficiency.
Experiments on several multimodal RAG benchmarks show improvements over multiple baseline methods.

**Compliance With Llm Reviewing Policy:**

Affirmed.

**Final Justification:**

Overall, I remain positive about this paper and support acceptance. The paper is original and meaningful in that it identifies an important mismatch between linear interaction histories and the actual reasoning structure of multimodal agentic RAG, and proposes a graph-based memory framework with graph-guided policy optimization to address it. I find the motivation clear, the method well designed, and the empirical improvements promising. While some concerns remain, especially regarding the expressive sufficiency of the DAG assumption, the rebuttal addressed most of my main questions and strengthened my confidence in the empirical validity of the work. Overall, I believe the paper makes a solid and valuable contribution.

**Key Questions For Authors:**

1. The proposed energy scoring combines structural information and semantic relevance. Could the authors provide additional intuition or empirical observations explaining why graph topology is a reliable indicator of evidence importance?

2. Graph-Guided Policy Optimization masks certain retrieval steps during optimization. How does this mechanism affect exploration during training, and could it potentially suppress useful but infrequent reasoning paths?

3. In long reasoning trajectories, does the graph memory grow unboundedly, or is there a mechanism for pruning or summarizing earlier nodes to control memory usage?

**Limitations:**

1. The proposed framework assumes that the reasoning process can be effectively represented as a directed acyclic graph. While this design is intuitive for multi-step retrieval tasks, it may not capture reasoning patterns that involve cycles, iterative refinement, or repeated evidence verification.

2. The token allocation strategy prioritizes evidence based on graph-derived energy scores. This approach implicitly assumes that graph topology correlates with semantic importance, which may not always hold for complex multimodal reasoning tasks.

**Strengths And Weaknesses:**

## Strengths

1. The motivation is clear and well presented. The figures are informative and help explain the framework effectively.

2. The paper identifies a meaningful challenge in multimodal RAG systems: visual evidence is often token-heavy yet semantically sparse. The mismatch between linear histories and the actual reasoning topology of agentic systems is an interesting observation.

3. The proposed Graph-Guided Policy Optimization provides a reasonable mechanism for improving credit assignment in multi-step reasoning trajectories and appears more principled than directly applying GRPO.

## Weaknesses

1. Mathematical notation and formulation are not rigorous. Eq.(12) contains a typo ("nax" instead of \max), and the expectation operator lacks proper conditioning. More importantly, Eq.(6) introduces the semantic priority $\hat{p}_{i,k}$ without a precise definition or computation procedure.

2. Experiments are limited to a single backbone architecture (Qwen3-VL). It is unclear whether the proposed framework generalizes to other multimodal models such as LLaVA or InternVL.

3. The evaluation relies heavily on a model-based metric using Qwen3-Max as the judge. The paper does not provide human evaluation or calibration analysis to validate the reliability of this metric.

4. The energy formulation (Eq.6–7) is heuristic and depends on hyperparameters such as $\lambda$ and $\gamma$, but the paper does not include sensitivity analysis or ablation studies.

5. The token allocation parameters in Eq.(8), including the top-K selection and token budget $S_{\text{total}}$, are not clearly specified, which affects reproducibility.

6. The computational overhead is not fully analyzed. Reporting only the average number of actions does not capture the additional cost of dynamic visual memory encoding and energy computation.

---

> ### Author Rebuttal · Authors · 2026-03-31
>
> Dear Reviewer sPDG,
>
> We sincerely appreciate your insightful review and the valuable insights you provided. Your recognition of our work's strengths has been a great encouragement to our team. We have tried our best to address all concerns. Please see our responses to each point below:
>
> >**W1: About the definitions of formulas and notations.**
>
> Following your valuable suggestion, we will add the definitions from Appendix to the main text to avoid possible concerns.
>
> >**W2: About the implementation on additional base models.**
>
> To further show the generalization ability of our approach, we applied our method to InternVL3‑8B, and the results are as follows:
>
> |Method|Retrieval Hit Rate|Overall Performance|
> |-|-|-|
> |ReAct|63.2|34.7|
> |Mem1|54.1|44.9|
> |VimRAG|**69.6**|**49.2**|
>
> The results show consistent improvements, indicating that VimRAG transfers well to different base models and has strong generalization ability.
>
> >**W3: About the consistency between LLM-based evaluators and human evaluations.**
>
> To address your concerns, we conducted experiments to demonstrate the consistency between human evaluations and model-based evaluators:
>
> |Evaluator|Pearson Correlation|
> |-|-|
> |Qwen3-Max vs. Human Expert|0.95|
> |Qwen3-Max vs. Gemini 3.1 Pro|0.97|
>
> The results show that current advanced LLMs show strong evaluation capabilities and align well with human experts.
>
> >**W4: About the sensitivity analysis and ablation studies on energy formulation.**
>
> We experimented with various parameters and conducted quantitative analysis during our research. The results are as follows:
>
> |λ|γ|Performance|
> |-|-|-|
> |0.1|0.2|49.9|
> |0.1|0.3|**50.1**|
> |0.3|0.3|49.7|
> |0.3|0.5|48.9|
>
> The results indicate that our setting with λ=0.1 and γ=0.3 achieves the highest overall performance, and our method also exhibits strong robustness.
>
> >**W5: About the experimental setup on Top-K and token budget.**
>
> (i) For the retriever, we set Top‑k to 3 across all baselines to ensure a fair comparison. (ii) We set the S_total=5\*256\*32\*32 (Appendix F).
>
> >**W6: About the computational overhead of energy computation.**
>
> The energy computation and dynamic visual memory encoding functions both have a complexity of O(n), where n denotes the number of nodes. The computation takes less than 1 ms per step, which is very small compared with the cost of model inference.
>
> >**Q1&L2: More explanations and empirical observations about relationship between graph topology and semantic importance.**
>
> We address the concerns in Q1 and L2 together, as they are related. The energy integrates semantic priority (assessed by the MLLM) with topological centrality (derived from the graph). We evaluate the relationship between topological positions and semantic relevance.
>
> |Node Out-Degree|Avg. Semantic Priority|Valuable Evidence Rate|
> |-|-|-|
> |1(Leaf)|2.8|22.4%|
> |2|3.5|44.2%|
> |≥3(Hub)|4.6|82.8%|
>
> The experimental results show that nodes with higher out-degrees are more likely to contain critical evidence and receive higher semantic scores, demonstrating that high-centrality nodes often serve as key bridges in multi‑hop reasoning.
>
> >**Q2: More analysis and clarification of the mask function in Graph-Guided Policy Optimization.**
>
> The masking mechanism improves sample quality without suppressing exploration. We simply set the gradients of masked steps to 0, applying no penalty or reward. To avoid off-policy training, the prompt context for updated steps still includes the masked steps.
>
> |Method|Invalid/Repeated Retrieval($\downarrow$)|Valuable Retrieval($\uparrow$)|
> |-|-|-|
> |w/o Graph Pruning|33.2%|42.1%|
> |w/ Graph Pruning (Ours)|**9.4%**|**72.6%**|
>
> Experiments show our optimization effectively produces valuable retrievals. Additionally, Fig. 6(b) shows our GGPO converges faster, confirming that it creates a consistent distribution and maintains exploration.
>
> >**Q3: About the growth of graph in long reasoning trajectories.**
>
> Thank you for this thoughtful observation. The graph memory does not grow without bound. We achieve this in two ways:
> + (i) a maximum step limit constrains the reasoning process (Algorithm 1 in the manuscript), and the model is prompted to generate a summary at the final step;
> + (ii) graph pruning during training removes redundant dead-end nodes, which naturally encourages more compact reasoning topologies during inference.
>
> >**L1: The performance of graph structures in various tasks.**
>
> We respectfully clarify that graph structures are inherently well-suited for these reasoning patterns. In fact, they offer fundamental advantages over linear structures.
>
> |Scenario|How graph works|
> |-|-|
> | Backtracking |New node can connect to any past node|
> |Iterative Refinement|Old node can be connected multiple times.|
> |Repeated Verification|Token re-allocation helps recall and boost past evidence|
>
> In conclusion, graph structures possess natural flexibility that allows them to handle a wide range of tasks.

---

> > ### Author Rebuttal · Reviewer_sPDG · 2026-04-01
> >
> > Thank you for the detailed rebuttal. The response does not fully address my main concern about the DAG assumption: while it explains why graph structures are more flexible than linear ones, it still does not clearly justify why a DAG is sufficient for reasoning patterns involving cycles, iterative refinement, and repeated verification, nor does it clarify the expressive limits of this design. That said, I still consider this to be a good paper overall.

---

> > > ### Author Response · Authors · 2026-04-02
> > >
> > > Dear Reviewer sPDG:
> > >
> > > Thank you so much for the recognition of our responses. We are glad to receive your positive feedback! Your review has been of great help to us in improving our manuscript. Thanks!
> > >
> > > We deeply value your time and expertise in reviewing our work. Here is a brief explanation of how the DAG handles these specific reasoning patterns. We will include more case studies and analyses in the revision to further illustrate this:
> > >
> > > | Reasoning Pattern | Examples in RAG task | How DAG Handles It |
> > > |:---|:---|:---|
> > > | Cycles | New step needs to circle back to a previously explored state. | Instead of forming a loop, a new node connects to historical nodes as a child, or creates a branching chain within the DAG. |
> > > | Iterative Refinement | New sub-queries need to reason with previous observations. | Nodes with multiple parents: A new node can connect to several historical nodes, merging past information without backward edges. |
> > > | Repeated Verification | New step or conclusion needs to repeatedly cross-check past evidence from earlier stages. | DAG-based Energy-driven memory: Keeps key evidence active for checking, eliminating the need to loop back or repeat retrieval. |
> > >
> > > Following your highly valuable suggestions, we will discuss the expressive limitations of the DAG in detail, such as the limited topological reasoning capabilities of smaller base models. We sincerely appreciate your insightful and constructive feedback, which has been incredibly valuable in helping us refine and strengthen our manuscript.
> > >
> > > Your support is truly helpful to us, and we are deeply grateful for your encouragement! **Many thanks for your constructive comments, time, and patience.**
> > >
> > > Best regards and thanks,
> > >
> > > The Authors

---

### Official Review · Reviewer_3B1n · 2026-03-12

**Soundness:** 3
**Presentation:** 4
**Significance:** 3
**Originality:** 3
**Overall Recommendation:** 4
**Confidence:** 3

**Summary:**

This paper proposes VimRAG, a multimodal RAG framework that organizes agent interaction history as a dynamic memory graph and further improves visual memory allocation and RL training with graph-guided designs. The problem is important, and the overall framework is interesting.

**Compliance With Llm Reviewing Policy:**

Affirmed.

**Final Justification:**

The authors have addressed my concerns, and I have no questions.

**Key Questions For Authors:**

1. Experimental fairness is not fully clear; the paper does not clearly specify whether all baselines use the same retrieval setup, action budget, and prompting protocol.
2. Some key design choices, especially in the visual memory allocation module, seem heuristic and are not sufficiently analyzed.
3. The RL optimization part is less convincing, since identifying valuable retrieval steps appears to require additional supervision.

**Limitations:**

see above

**Strengths And Weaknesses:**

1. The paper studies an important problem: multimodal agentic RAG with long-horizon reasoning and large visual context.
2. The overall framework is interesting and reasonably well motivated, combining graph memory, visual memory allocation, and graph-guided policy optimization in a unified design.
3. Experimental results are promising, with consistent gains over baselines on multiple benchmarks.
4. The paper includes ablation and efficiency analysis, which improves empirical completeness.

---

> ### Author Rebuttal · Authors · 2026-03-31
>
> Dear Reviewer 3B1n,
>
> We sincerely appreciate your insightful review and the valuable insights you provided. Your recognition of our work's strengths has been a great encouragement to our team. We have tried our best to address all concerns. Please see our responses to each point below:
>
> > **Q1: Clarification about the experimental details and the baseline setting.**
>
> Thank you for your highly valuable suggestion. Following your advice, we clarify our experimental settings to highlight the fairness of our experiments.
>
> The baselines and our VimRAG employ the same retrieval setup (same search engine over the ~200k-item corpus, in Appendix B), identical action budgets (same $T_{\max}$ and action protocols), and standardized prompting (Appendix D & J), as summarized below:
>
> | Aspect | Details for fair comparison  | Paragraph in Manuscript |
> | --- | --- | --- |
> | Retriever | All methods use the same search engine over the same 200k-item corpus with unified multimodal preprocessing. | Section 4.1, Appendix B |
> | Action Budget | The same step limits and action protocol across all methods. | Section 4.1, Appendix B |
> | Prompting Protocol | All methods use officially released prompts with the same tool protocols. | Appendix D & J |
>
> > **Q2: More discussion about the motivation for designing the visual memory module.**
>
> Thank you so much for raising this thoughtful concern. We would like to clarify that the visual memory allocation mechanism is inspired by the empirical insights from our pilot study (Sec. 2.2 & 2.3), which systematically investigated the trade-offs between compression and information preservation.
>
> As illustrated in the $1_{st}$ pilot study (Sec. 2.2 & Fig. 2), the graph-based paradigm significantly reduces invalid retrieval actions compared to summary-based methods, motivating our use of graph structures to preserve agent reasoning states. Furthermore, the $2_{nd}$ pilot study (Sec. 2.3 & Tab. 1) reveals that semantically related visual memory achieves the best performance (58.2%/43.7%) by selectively retaining critical visual tokens. Our energy formulation implements this selective retention via intrinsic semantic scores and topology-aware propagation.
>
> Based on the empirical findings above, we design the visual memory module as a high-pass filter.
>
> | Type | Node Characteristics in Graph | Energy Dynamics | Outcome | Example |
> | --- | --- | --- | --- | --- |
> | Low energy | Old and Isolated (no children)| Decays to near-zero | Compressed/dropped | Dead-end steps or irrelevant outdated nodes.|
> | High energy | New observations or have active descendants| Child node feedback boosts energy | High-resolution retention | Key definitions or evidence|
>
> Through qualitative and quantitative analysis, we designed the visual memory module, which implements an optimal evidence selector that preserves structurally and semantically critical evidence while discarding low‑value noise.
>
> > **Q3: Regarding the potential additional supervision for retrieval actions in RL training.**
>
> We appreciate this insightful inquiry. Our fine-grained supervision is strictly zero-cost because it exploits how RAG datasets are inherently constructed: queries are generated from specific source documents, making the query-to-source mapping a natural byproduct of data creation (mentioned in Appendix A). The dataset example is as follows:
>
> ```json
> {
>   "qid": "unique_query_id",
>   "query": "query_content",
>   "gt": "golden_answer",
>   "file_name": "file_name_of_text_file/image_page/video_clip"
> }
> ```
>
> During RL training, if the agent retrieves the file related to the query, it is considered a valuable retrieval step. This allows us to perform fine-grained credit assignment without introducing any additional human annotation or computational overhead.

---

> > ### Author Rebuttal · Reviewer_3B1n · 2026-04-01
> >
> > I have read the rebuttal, and most of my concerns are addressed. I will keep my score to vote accept.

---

> > > ### Author Response · Authors · 2026-04-02
> > >
> > > Dear Reviewer 3B1n:
> > >
> > > We sincerely appreciate your insightful review. Your positive feedback and support for our work are truly encouraging to us.
> > >
> > > We are pleased to know that our rebuttal has fully addressed your concerns. We are deeply grateful for the professional guidance you provided throughout the review process.
> > >
> > > **Many thanks for your constructive comments, time, and patience.**
> > >
> > > Best regards and thanks,
> > >
> > > The Authors

---

### Decision · Program_Chairs · 2026-04-30

**Decision:**

Accept (regular)

**Comment:**

After reviewing the paper, the reviewers’ comments, and the authors’ rebuttal, I recommend Accept. The rebuttal process was constructive and thorough. The authors addressed the key concerns raised by the reviewers, including clarifying experimental setups, providing additional results on alternative backbone models (e.g., InternVL3-8B), adding sensitivity analyses for key hyper-parameters, demonstrating consistency between model-based and human evaluations, and explaining the design rationale behind the graph-based memory and energy formulation. While one reviewer raised a remaining question about the sufficiency of the DAG assumption for cyclic reasoning patterns, the authors provided a reasonable justification and committed to further discussion in the revision. Overall, the authors responded professionally and substantively, and the manuscript has been strengthened accordingly.